# Evaluation of water flux predictive models developed using eddy
# covariance observations and machine learning: a meta-analysis
Haiyang Shi[1,2,4,5], Geping Luo[1,2,3,5], Olaf Hellwich[6], Mingjuan Xie[1,2,4,5], Chen Zhang[1,2], Yu Zhang[1,2], Yuangang
Wang[1,2], Xiuliang Yuan[1], Xiaofei Ma[1], Wenqiang Zhang[1,2,4,5], Alishir Kurban[1,2,3,5], Philippe De Maeyer[1,2,4,5] and
Tim Van de Voorde[4,5]
[1] State Key Laboratory of Desert and Oasis Ecology, Xinjiang Institute of Ecology and Geography, Chinese
Academy of Sciences, Urumqi, Xinjiang, 830011, China.
[2] University of Chinese Academy of Sciences, 19 (A) Yuquan Road, Beijing, 100049, China.
[3] Research Centre for Ecology and Environment of Central Asia, Chinese Academy of Sciences, Urumqi, China.
[4] Department of Geography, Ghent University, Ghent 9000, Belgium.
[5] Sino-Belgian Joint Laboratory of Geo-Information, Ghent, Belgium and Urumqi, China.
[6] Department of Computer Vision & Remote Sensing, Technische Universität Berlin, 10587 Berlin, Germany.
***Correspondence to*: Geping Luo (luogp@ms.xjb.ac.cn) and Olaf Hellwich (olaf.hellwich@tu-berlin.de)**
Submitted to *Hydrology and Earth System Sciences*

**Abstract.**

With the rapid accumulation of water flux observations from global eddy-covariance flux sites, many studies have used data-driven approaches to model water fluxes with various predictors and machine learning algorithms used. However, systematic evaluation of such models is still limited. We therefore performed a meta-analysis of 32 such studies, derived 139 model records, and evaluated the impact of various features on model accuracy throughout the modeling flow. SVM (average R-squared = 0.82) and RF (average R-squared = 0.81) outperformed over evaluated algorithms with sufficient sample size in both cross-study and intra-study (with the same data) comparisons. The average accuracy of the model applied to arid regions is higher than in other climate types. The average accuracy of the model was slightly lower for forest sites (average R-squared = 0.76) than for croplands and grasslands (average R-squared = 0.8 and 0.79), but higher than for shrubland sites (average R-squared = 0.67). Using Rn/Rs, precipitation, Ta, and FAPAR improved the model accuracy. The combined use of Ta and Rn/Rs is very effective especially in forests, while in grasslands the combination of Ws and Rn/Rs is also effective. Random cross-validation showed higher model accuracy than spatial cross-validation and temporal cross-validation, but spatial cross-validation is more important in spatial extrapolation. The findings of this study are promising to guide future research on such machine learning-based modeling.

## 1 Introduction

Evapotranspiration (ET) is one of the most important components of the water cycle in terrestrial ecosystems. It also represents the key variable in linking ecosystem functioning, carbon and climate feedbacks, agricultural management, and water resources (Fisher et al., 2017). The quantification of ET for regional, continents, or the globe can improve our understanding of the water, heat, and carbon interactions, which is important for global change research (Xu et al., 2018). Information on ET has been used in many fields, including, but not limited to, droughts and heatwaves (Miralles et al., 2014), regional water balance closures (Chen et al., 2014; Sahoo et al., 2011), agricultural management (Allen et al., 2011), water resources management (Anderson et al., 2012), biodiversity patterns (Gaston, 2000). In addition, accurate large-scale and long-time series ET prediction at high spatial and temporal resolution has been of great interest  (Fisher et al., 2017).

Currently, there are three main approaches for simulation and spatial and temporal prediction of ET: (i) physical models based on remote sensing such as surface energy balance models (Minacapilli et al., 2009; Wagle et al., 2017), Penman-Monteith equation (Mu et al., 2011; Zhang et al., 2010), Priestley-Taylor equation (Miralles et al., 2011); (ii) process-based land surface models, biogeochemical models and hydrological models (Barman et al., 2014; Pan et al., 2015; Sándor et al., 2016; Chen et al., 2019); and (iii) the observation-based machine learning modeling approach with in situ eddy covariance (EC) observations of water flux (Jung et al., 2011; Li et al., 2018; Van Wijk and Bouten, 1999; Xie et al., 2021; Xu et al., 2018; Yang et al., 2006; Zhang et al., 2021). For remote sensing-based physical models and process-based land surface models, some physical processes have not been well characterized due to the lack of understanding of the detailed mechanisms influencing ET under different environmental conditions. For example, the inaccurate representation and estimation of stomatal conductance (Li et al., 2019) and the linearization (McColl, 2020) of the Clausius-Clapeyron relation in the Penman-Monteith equation may introduce both empirical and conceptual errors into estimates of ET. Limited by

complicated assumptions and model parametrizations, these process-based models face challenges in the
accuracy of their ET estimations over heterogeneous landscapes (Pan et al., 2020; Zhang et al., 2021).
Therefore, many researchers have used data-driven approaches for the simulation and prediction of ET with the
accumulation of a large volume of measured observational data of water fluxes in the past decades. Various
machine learning models have been developed to simulate water fluxes at the flux site scale. Besides, various
predictor variables (e.g., meteorological factors, vegetation conditions, and moisture supply conditions) have
been incorporated into such models for upscaling (Fang et al., 2020; Jung et al., 2009) of water flux to a larger
scale or understanding the driving mechanisms with the variable importance analysis performed in such models.

However, to date, the systematic assessment of the uncertainty in the processes of water flux prediction models
constructed using the machine learning approach is limited. Although considerable effort has been invested in
improving the accuracy of such prediction models, our understanding of the expected accuracy of such models
under different conditions is still limited. It is still not easy for us to give the general guidelines for selecting
appropriate predictor variables and models. Questions such as 'Which predictor variables are the best in water
flux simulations?' and 'How to improve the prediction accuracy of water flux effectively?' etc. still confuse the
researchers in the field. Therefore, we should synthesize the findings from published studies to determine which
predictor variables, machine learning models, and other features can significantly improve the prediction
accuracy of water flux. Also, we are interested in understanding under which specific conditions they are more
effective.

A variety of features control the accuracy of such models, including the predictor variables used, the inherent
heterogeneity within the dataset, the plant functional type (PFT) of the flux sites, the method of model
construction and validation, and the algorithm chosen:
a)   Predictor variables used: Compared to process-based models, the data used may have a more significant

impact on the final model performance in data-driven models. Various biophysical covariates and other

environmental factors have been used for the simulation and prediction of water fluxes. The most

commonly used factors include mainly precipitation (Prec), air temperature (Ta), wind speed (Ws), net/sun

radiation (Rn/Rs), soil temperature (Ts), soil texture, vapor-pressure deficit (VPD), the fraction of absorbed

photosynthetically active radiation (FAPAR), vegetation index (e.g., Normalized Difference Vegetation

Index (NDVI), Enhanced Vegetation Index (EVI)), Leaf area index (LAI), and carbon fluxes (e.g., Gross

Primary Productivity (GPP)). These used predictor variables and their complex interactions drive the

fluctuations and variability of water fluxes. They affect the accuracy of water flux simulations in two ways:

their actual impact on water fluxes at the process-based level and their spatio-temporal resolution and

inherent accuracy. The relationship between water fluxes and these variables at the process-based driving

mechanism level is very different under different PFTs, different climate types, and different

hydrometeorological conditions. For example, in irrigated croplands in arid regions, water fluxes may be

highly correlated with irrigation practices, and thus soil moisture may be a very important predictor

variable, and its importance may be significantly higher than in other PFTs. And in models that incorporate

data from multiple PFTs, some variables that play important roles in multiple PFTs may have higher

importance. In terms of data spatial and temporal resolution, the data for these predictor variables may have

different scales. In terms of spatial resolution, meteorological observations such as precipitation and air temperature are at the flux site scale, while factors extracted from satellite remote sensing and reanalysis climate datasets cover a much larger spatial scale (i.e. the grid-scale). This leads to considerable differences in the degree of spatial match between different variables and the site scale EC observations (approximately 100 m x 100 m). It is therefore difficult for some variables to be fairly compared in the subsequent importance analysis of driving factors. In terms of temporal resolution, the importance of predictor variables with different temporal resolutions may be variable for models with different time scales (e.g., half-hourly, daily, and monthly models). For example, the daily or 8-day NDVI data based on MODIS satellite imagery may better capture the temporal dynamics of water fluxes concerning vegetation growth than the 16-daily NDVI data derived from Landsat images. Besides, data on non-temporal dynamic variables such as soil texture cannot explain temporal variability in water fluxes in the data-driven simulations, although soil texture may be important in the interpretation of the actual driving mechanisms of ET (which may need to be quantified in detail in ET simulations by process-based models). In addition, some inherent accuracy issues (e.g., remote sensing-based NDVI may not be effective at high values) of the predictors may propagate into the consequent machine learning models, thus affecting the modeling and our understanding of its importance. Therefore, it is necessary to consider the spatial and temporal resolution of the data and their inherent accuracy for the predictors used in different studies in the systematic evaluation of data-driven water flux simulations.

b) The heterogeneity of the dataset and model validation: the volume and inherent spatiotemporal heterogeneity of the training dataset (with more variability and extremes incorporated) may affect model accuracy. Typically, training data with larger regions, multiple sites, multiple PFTs, and longer year spans may have a higher degree of imbalance (Kaur et al., 2019; Van Hulse et al., 2007; Virkkala et al., 2021; Zeng et al., 2020). And in machine learning, in general, modeling with unbalanced data (with significant differences in the distribution between the training and validation sets) may result in lower model accuracy. Currently, the most common ways of model validation include spatial, temporal, and random cross-validation. Spatial validation is mainly to evaluate the ability of the model to be applied in different regions or flux sites with different PFT types, and one of the common methods is 'leave one site out' (Fang et al., 2020; Papale et al., 2015; Zhang et al., 2021). If the data of the site left out for validation differs significantly from the distribution of the training data set, the expected accuracy of the model applied at that site may be low because the trained model may not capture the specific and local relationships between the water flux and the various predictor variables at that site. For temporal validation, to assess the ability of the models to adapt to the interannual variability, typically some years of data are used for training and the remaining years for model validation (Lu and Zhuang, 2010). If a year with extreme climate is used for validation, the accuracy may be low because the training dataset may not contain such extreme climate conditions. In the case of PFTs that are significantly affected by human activities, such as cropland, the possible different crops grown and different land use practices (e.g., irrigation) across years can also lead to low accuracy in temporal validation.

c) Various machine learning algorithms: Some machine learning algorithms may have specific advantages when applied to model the relationships between water fluxes and covariates. For example, neural networks may have an advantage in nonlinear fitting, while random forests can avoid serious overfitting problems.

135       However, which algorithm is better overall in different situations (i.e. applied to different data sets)? Which

algorithm is generally more accurate than the others when using the same data set? A comprehensive

evaluation is important.


Therefore, to systematically and comprehensively assess the impact of various features in such modeling, we
perform a meta-analysis of published water flux simulation studies that combine the flux site water flux
observations, various predictors, and machine learning. The accuracy of model records collected from the
literature was linked with various model features to assess the impacts of predictor data types, algorithms, and
other features on model accuracy. The findings of this study may be promising to improve our understanding of
the impact of various features of the models to guide future research on such machine learning-based modeling.
**2 Methodology**
**2.1 Protocol for selecting the sample of articles**
We applied a general query (on December 1st, 2021) on title, abstract, and keywords to include articles with the
"OR" operator applied among expressions (Table 1) in the Scopus database. Preferred Reporting Items for
Systematic Reviews and Meta-Analyses (PRISMA) (Moher et al., 2009) are followed when filtering the papers.
We first excluded articles that obviously did not fit the topic of this study based on the abstract, and then
performed the article screening with the full-text reading.

The inclusion of articles follows the following criteria:
a)   Articles were filtered for those with water fluxes (or latent heat) simulated.
b)   The water flux or latent heat observations used in the prediction models should be from the eddy-

covariance flux measurements.

c)   Articles focusing only on gap-filling (Hui et al., 2004) techniques (i.e., the objective was not simulation

and extrapolation of water fluxes using machine learning) were excluded.

d)   Only articles that used multivariate regression (with the number of covariates greater than or equal to 3)

were included.

e)   The determination coefficient (R-squared) of the validation step should be reported as the metric of model

performance (Shi et al., 2021; Tramontana et al., 2016; Zeng et al., 2020) in the articles.

f)   The articles should be published in English-language journals.

Although RMSE is also often used for model accuracy assessment, its dependence on the magnitude of water
flux values makes it difficult to use for fair comparisons between studies. For example, due to the difference in
the range of ET values, models developed from flux stations in dry grasslands will typically have lower RMSE
than models developed by flux stations based on forests in humid regions. Therefore, RMSE may not be a good
metric for cross-study comparisons in this meta-analysis.

Table 1. Article search: '[A1 OR A2 OR A3...] AND [B1 OR B2 OR B3...] AND [C1 OR C2 OR C3 OR C4...]'

| ID | A | B | C |
|---|---|---|---|
| 1 | Water flux | Eddy covariance | Machine learning |
| 2 | Evapotranspiration | Flux tower | Support Vector |
| 3 | Latent heat | Flux site | Neural Network |
| 4 | | | Random Forest |


### 2.2 Features of the prediction processes evaluated

The various features (Table 2) involved in the water flux modeling framework (Fig. 1) include the PFTs of the
sites, the predictors used, the machine learning algorithms, the validation methods, and other features. Each
model for which R-squared is reported is treated as a data record. If multiple algorithms were applied to the
same dataset, then multiple records were extracted. Models using different data or features are also recorded as
multiple records.

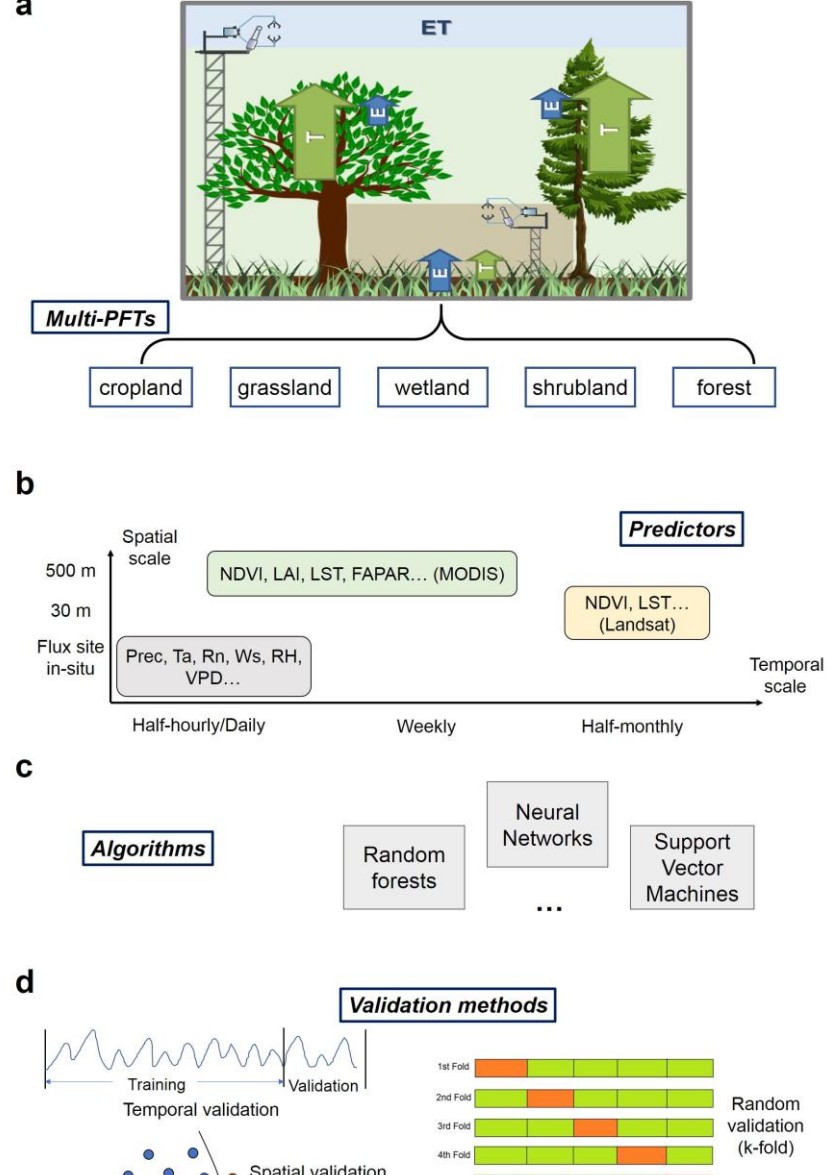


Figure 1. Features of the machine learning-based water flux prediction process. (a) the eddy-covariance-based water flux observations of various plant function types (PFTs), modified from Paul-Limoges et al., 2020. ET, evapotranspiration. E, evaporation. T, transpiration. (b) Predictors and their spatial and temporal resolution. (c) The machine learning algorithms used for the modeling, such as neural networks, random forests, etc. (d) The model validation methods used including the spatial, temporal, and random cross-validations.

185

Table 2. Description of information extracted from the included papers.

| Field | Definition & Categories adopted | Harmonization |
|---|---|---|
| Climate | Climate zones of the study location derived from the Köppen climate classification (Peel et al., 2007) | |
| Plant functional type (PFT) | PFT of the flux sites: 1-forest, 2-grassland, 3-cropland, 4-wetland, 5-shrubland, 6-savannah, and multi-PFTs | The categorization is based on the descriptions in the article. For example, cropland for various crops is classified |

| | | |
|---|---|---|
| | | as 'cropland', and both woody savannah and savannah are classified as 'savannah'. |
| Location | More precise location (with the latitude and longitude of the center of the studied sites): latitude, longitude | |
| Algorithms | Random Forests (RF), Multiple Linear Regressions (MLR), Artificial Neural Networks (ANN), Support Vector Machines (SVM), Cubist, model tree ensembles (MTE), K-nearest neighbors (KNN), long short-term memory (LSTM), gradient boosting regression tree (GBRT), extra tree regressor (ETR), Gaussian process regression (GPR), Bayesian model averaging (BMA), extreme learning machine (ELM), and deep belief network (DBN) | Various model algorithms with parameter optimization or other improvements are categorized as their algorithm family. For example, various improved models of RF algorithms are classified as RF, rather than as another algorithm family. |
| Sites number | Number of the flux sites used | |
| Spatial scale | Area representatively covered by the flux sites: local (less than 100 x 100 km), regional, global (continent-scale and global scale) | The spatial scale is roughly categorized based on the area covered by the site. The model is classified as 'global' only when the spatial extent reaches the continental scale. |
| Temporal scale | The temporal scale of the model: half-hourly, hourly, daily, 4-daily, 8-daily, monthly, seasonally (i.e., 0.02, 0.04, 1, 4, 8, 30, 90 days) | Models with a temporal scale greater than one month and less than one year are classified as seasonal scale models. |
| Year span | The span of years of the flux data used | Year span is calculated as the span from the earliest to the latest year of available flux data. |
| Site year | Describe the volume of total flux data with the number of sites and years aggregated. | |
| Cross-validation | Describe the chosen method of cross-validation: Spatial (e.g., 'leave one site out'), temporal (e.g., 'leave one year out'), random (e.g., 'k-fold') | |
| Training/validation | Describe the ratio of the data volume in the training and validation sets. | In spatial validation, this ratio is represented by the ratio of the number of sites used for training to the number of sites used for validation. In temporal validation, this is represented by the ratio of the span of time periods used for training to the span of time periods used for validation. |
| Satellite images | Describe the source of satellite images used to derive NDVI, EVI, LAI, LST, etc: Landsat, MODIS, AVHRR | |
| Biophysical predictors | LAI, NDVI/EVI, the fraction of absorbed photosynthetically active radiation/photosynthetically active radiation (FAPAR/PAR), leaf area index (LAI), Carbon fluxes (CF) including NEE/GPP, etc. | The predictor variables of different measurement methods are categorized according to their definitions. For example, both using the NDVI calculated based on satellite remote |

| | | sensing bands and in situ measurements were classified as the use of 'NDVI'. |
|---|---|---|
| Meteorological variables | precipitation (Prec), net radiation/solar radiation (Rn/Rs), air temperature (Ta), vapour-pressure deficit (VPD), relative humidity (RH) , etc. | The way meteorological data are measured is not differentiated. For example, both using Ta from reanalysis data and Ta measured at flux sites were classified as the use of Ta. |
| Ancillary data | Describe the ancillary variables used: soil texture, terrain (DEM), soil moisture/land surface water index (SM/LSWI), etc. | Both the use of in situ measured soil moisture and the use of remote sensing-based LSWI was classified as using surface moisture-related indicators SM/LSWI. |
| Accuracy measure | Accuracy measure used to assess the model performance: R-squared (in the validation phase) | |

187

## 3 Results

### 3.1 Articles included in the meta-analysis

A total of 32 articles (Table S1) containing a total of 139 model records were included. The geographical scope of these articles was mainly Europe, North America, and China (Fig. 2).

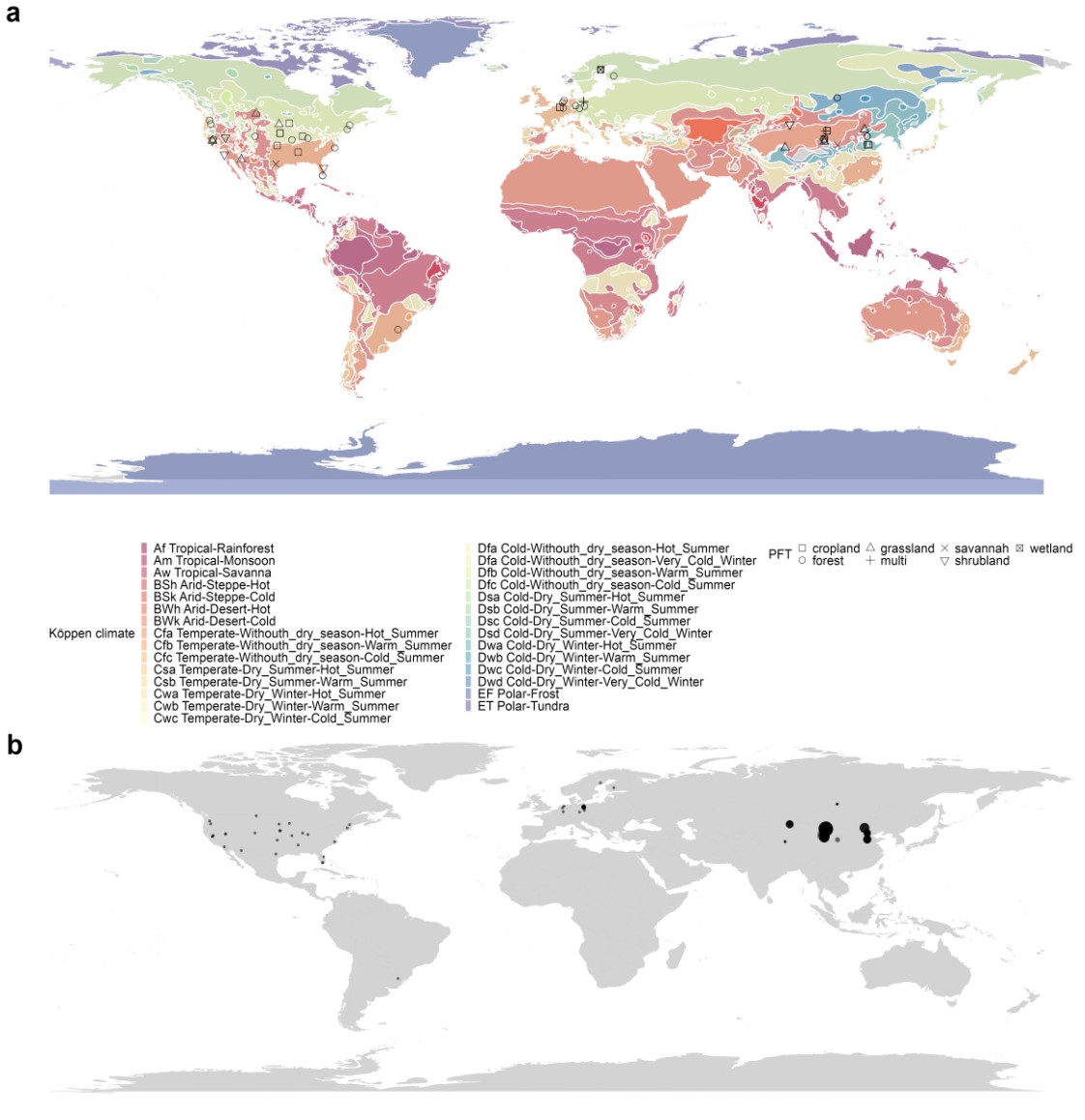

a

b

Köppen climate

| Af Tropical-Rainforest | Dfa Cold-Withouth_dry_season-Hot_Summer | PFT □ cropland △ grassland × savannah ⊠ wetland |
| Am Tropical-Monsoon | Dfa Cold-Withouth_dry_season-Very_Cold_Winter | ○ forest + multi ▽ shrubland |

Af Tropical-Rainforest
Am Tropical-Monsoon
Aw Tropical-Savanna
BSh Arid-Steppe-Hot
BSk Arid-Steppe-Cold
BWh Arid-Desert-Hot
BWk Arid-Desert-Cold
Cfa Temperate-Withouth_dry_season-Hot_Summer
Cfb Temperate-Withouth_dry_season-Warm_Summer
Cfc Temperate-Withouth_dry_season-Cold_Summer
Csa Temperate-Dry_Summer-Hot_Summer
Csb Temperate-Dry_Summer-Warm_Summer
Cwa Temperate-Dry_Winter-Hot_Summer
Cwb Temperate-Dry_Winter-Warm_Summer
Cwc Temperate-Dry_Winter-Cold_Summer

Dfa Cold-Withouth_dry_season-Hot_Summer
Dfa Cold-Withouth_dry_season-Very_Cold_Winter
Dfb Cold-Withouth_dry_season-Warm_Summer
Dfc Cold-Withouth_dry_season-Cold_Summer
Dsa Cold-Dry_Summer-Hot_Summer
Dsb Cold-Dry_Summer-Warm_Summer
Dsc Cold-Dry_Summer-Cold_Summer
Dsd Cold-Dry_Summer-Very_Cold_Winter
Dwa Cold-Dry_Winter-Hot_Summer
Dwb Cold-Dry_Winter-Warm_Summer
Dwc Cold-Dry_Winter-Cold_Summer
Dwd Cold-Dry_Winter-Very_Cold_Winter
EF Polar-Frost
ET Polar-Tundra

number of sites · 1 • 5 ● 10 ● 20

Figure 2. Location of the included studies in the meta-analysis. (a) PFTs and the climate zones (from Köppen climate classification) of these studies and (b) the number of flux sites included in each study. Global and continental-scale studies (e.g., models developed based on FLUXNET of the global scale) are not shown on the map due to the difficulty of identifying specific locations.

**3.2 The formal Meta-analysis**

**3.2.1 Algorithms**

SVM and RF outperformed (Fig. 3a) across studies (better than other algorithms with sufficient sample size in Fig. 3a such as ANN). These three machine learning algorithms (i.e., ANN, SVM, RF) were significantly more accurate than the traditional MLR. Other algorithms such as MTE, ELM, Cubist, etc. also correspond to high accuracy, but with limited evidence sample size (Fig. 3a). In the internal comparison (different algorithms applied to the same data set) in single studies, we also find that SVM and RF were slightly more accurate than ANN (Fig. 3b), and all these three (i.e., ANN, SVM, RF) are considerably more accurate than MLR. Overall,

SVM and RF have shown higher accuracy in water flux simulations in both inter and intra-study comparisons
with sufficient sample size as evidence.

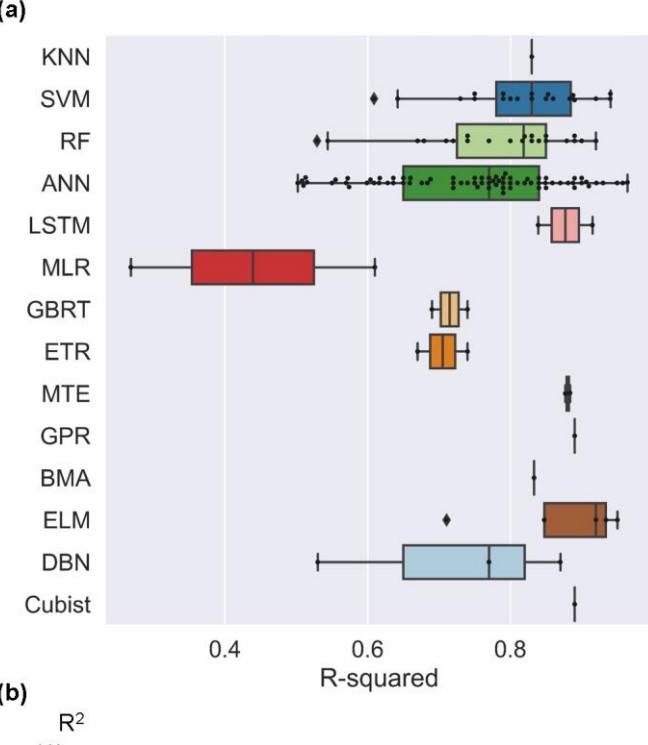

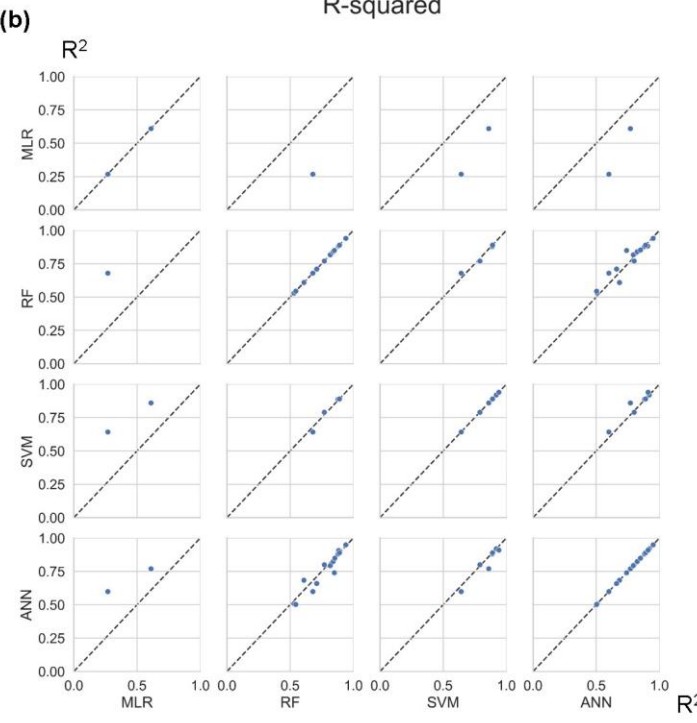


Figure 3. Model accuracy (R-squared) using various algorithms across studies (a) and internal comparisons of
selected pairs of algorithms within studies (b). Algorithms: Random Forests (RF), Multiple Linear Regressions
(MLR), Artificial Neural Networks (ANN), Support Vector Machines (SVM), Bayesian model averaging
(BMA), Cubist, model tree ensembles (MTE), gradient boosting regression tree (GBRT), extra tree regressor
(ETR), K-nearest neighbors (KNN), long short-term memory (LSTM), Gaussian process regression (GPR),
extreme learning machine (ELM), and deep belief network (DBN).

### 3.2.2 Climate types and PFTs

We found higher average model accuracy in arid climate zones (Fig. 4a), such as the Cold semi-arid (steppe) climate (BSk) and Cold desert climate (BWk). Most of these studies were located in northwest China and the western USA. It may be caused by the simpler relationship between water fluxes and biophysical covariates in arid regions. In arid zones, due to the high potential ET, the variability in the actual ET may be largely explained by water availability (moisture supply) and vegetation change with the effect of variability in thermal conditions reduced. As for the various PFTs, the average model accuracy was slightly lower for forest types than for cropland and grassland types (Fig. 4b). The lowest average accuracy was found for shrub sites, which may be related to the difficulty of the remote sensing-based vegetation index (e.g., NDVI) to quantify the physiological and ecological conditions of shrubs (Zeng et al., 2022), and the heterogeneity of the spatial distribution of shrubs within the EC observation area may also cause difficulties in capturing their relationships with biophysical variables. We also found high model accuracy for the wetland type, although records as evidence to support this finding may be limited. Compared to other PFTs, the more steady and adequate water availability in the wetland type may make the variations of water fluxes less explained by other biophysical covariates.

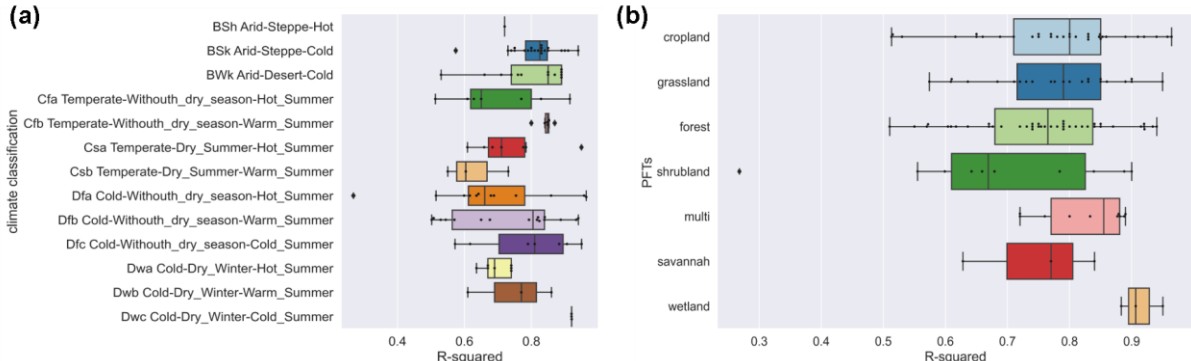

Figure 4. Differences in model accuracy (R-squared) of (a) various climate zones (classified by Köppen climate classification) across studies and (b) PFTs. BSh, Hot semi-arid (steppe) climate. BSk, Cold semi-arid (steppe) climate. BWk, Cold desert climate. Cfa, Humid subtropical climate. Cfb, Temperate oceanic climate. Csa, Hot-summer Mediterranean climate. Csb, Warm-summer Mediterranean climate. Dfa, Hot-summer humid continental climate. Dfb, Warm-summer humid continental climate. Dfc, Subarctic climate. Dwa, Monsoon-influenced hot-summer humid continental climate. Dwb, Monsoon-influenced warm-summer humid continental climate. Dwc, Monsoon-influenced subarctic climate.

### 3.3.3 Predictors and their combinations

On one hand, for the effects of individual predictors, the use of Rn/Rs, Prec, Ta, and FAPAR improved the accuracy of the model (Fig. S1). This pattern partially changed in the different PFTs. In the forest sites, the accuracy of the models with Rn/Rs and Ta used was higher than that of the models with Rn/Rs and Ta not used. For the grassland sites, the use of Ws, FAPAR, Prec, and Rn/Rs improved the model accuracy. For the cropland sites, Ta and FAPAR were more important for improving the model accuracy.

On the other hand, the evaluation of the effect of individual predictors on model accuracy is not necessarily reliable because some predictor variables are used together (e.g., the high model accuracy corresponding to a particular variable may be because it is often used together with another variable that plays the dominant role in

improving accuracy). Therefore, we tested for independence between the use of variables and assessed the effect
of the combination of variables on model accuracy. We calculated the correlation matrix (Fig. S2) between the
use of various predictors (not used is set as 0 and used is set as 1). We found there was a dependence between
the use of some predictors, the use of NDVI/EVI, LAI, and SM was significantly negatively correlated with the
use of Rn/Rs and Ta (Fig. S2). It indicated that many of the models that used Rn/Rs and Ta did not use
NDVI/EVI, LAI, and SM, and the models that used NDVI/EVI, LAI, and SM also happened to not use Rn/Rs
and Ta. Given this dependence, we evaluated the effect of the combination of variables on the model accuracy
(Fig. 5). In Fig. 5, the three variable combinations on the left side are mainly meteorological variables while the
three variable combinations on the right side are mainly vegetation-related variables based on remote sensing
(e.g., NDVI, EVI, LAI, LSWI). We found that, overall, the accuracy of the models using only meteorological
variable combinations was higher than that of the models using only remote sensing-based vegetation-related
variables. It demonstrated the importance of using meteorological variables in machine learning-based ET
prediction (probably especially for models with small time scales such as hourly scale, and daily scale). For
example, in the forest type, the combination of Ta and Rn/Rs is very effective compared to using only remote
sensing-based vegetation index variable combinations. The combination of Ta and Rn/Rs is also effective in the
grassland and cropland types. The combination of Ws and Rn/Rs played an important role in the grassland type
for improving model accuracy. Despite this, it does not negate the positive role of remote sensing-based
vegetation-related variables in ET prediction. This effectiveness can be dependent on the time scale of the model
as well as the PFTs. In models with large time scales (monthly scale, seasonal scale) and PFTs in which ET is
sensitive to vegetation dynamics, remote sensing-based vegetation-related variables may also be of high
importance.

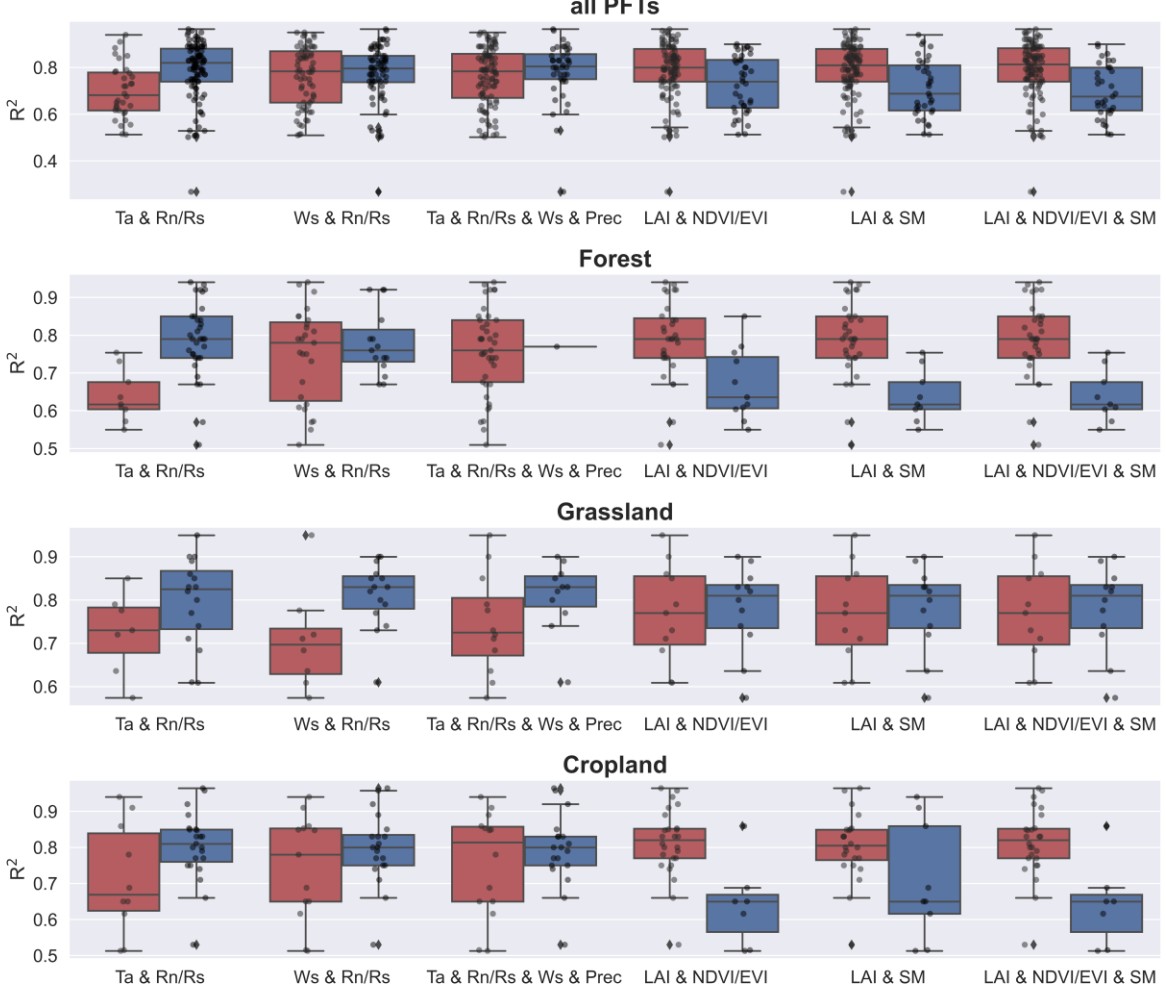

Figure 5. Effects of combinations of predictor variables on model accuracy in various PFTs (all data, forest, grassland, and cropland). Dark blue boxes indicate that the predictors were together used in the model (e.g., for 'Ta & Rn/Rs', the dark blue box represents Ta and Rn/Rs were together used in the model), while dark red boxes indicate the other conditions (i.e., the combination was not used). Predictors: precipitation (Prec), soil moisture/remote sensing-based land surface water index (SM), net radiation/solar radiation (Rn/Rs), enhanced vegetation index (EVI), air temperature (Ta), leaf area index (LAI), Normalized Difference Vegetation Index/Enhanced Vegetation Index (NDVI/EVI).

### 3.3.4 Other model features

We also evaluated the impact of some other features on accuracy. The differences in accuracy of models with different spatial scales, year spans, number of sites, and volume of data (Fig. 6) appear to be insignificant. This seems to be related to the fact that in large-scale water flux simulations, the sites of similar PFTs are selected such as for modeling multiple forest sites across Europe (Van Wijk and Bouten, 1999) which focus on 'forest' and multiple grassland sites across arid northern China (Xie et al., 2021; Zhang et al., 2021) which focus on 'grassland', rather than mixing different PFT types to train models as the way in machine learning modeling of carbon fluxes (Zeng et al., 2020). In terms of the time scales of the models, the 4-day, 8-day, and monthly scales appear to correspond to higher accuracy compared to the half-hourly and daily scales. The higher the ratio of the volume of data in the training and validation sets, the higher the model accuracy. Compared to the models using

Landsat data, the models using MODIS data showed slightly higher accuracy probably due to the advantage of MODIS data in capturing the temporal dynamics of biophysical covariates. There were significant differences in the accuracy of the models using different cross-validation methods, with the models using random cross-validation showing higher accuracy than those using temporal cross-validation. This suggests that interannual variability may have a high impact on the models in water flux simulations. The driving mechanism of ET may vary significantly across years, and the inclusion of some extreme climatic conditions in the training set may be important for model accuracy and robustness.

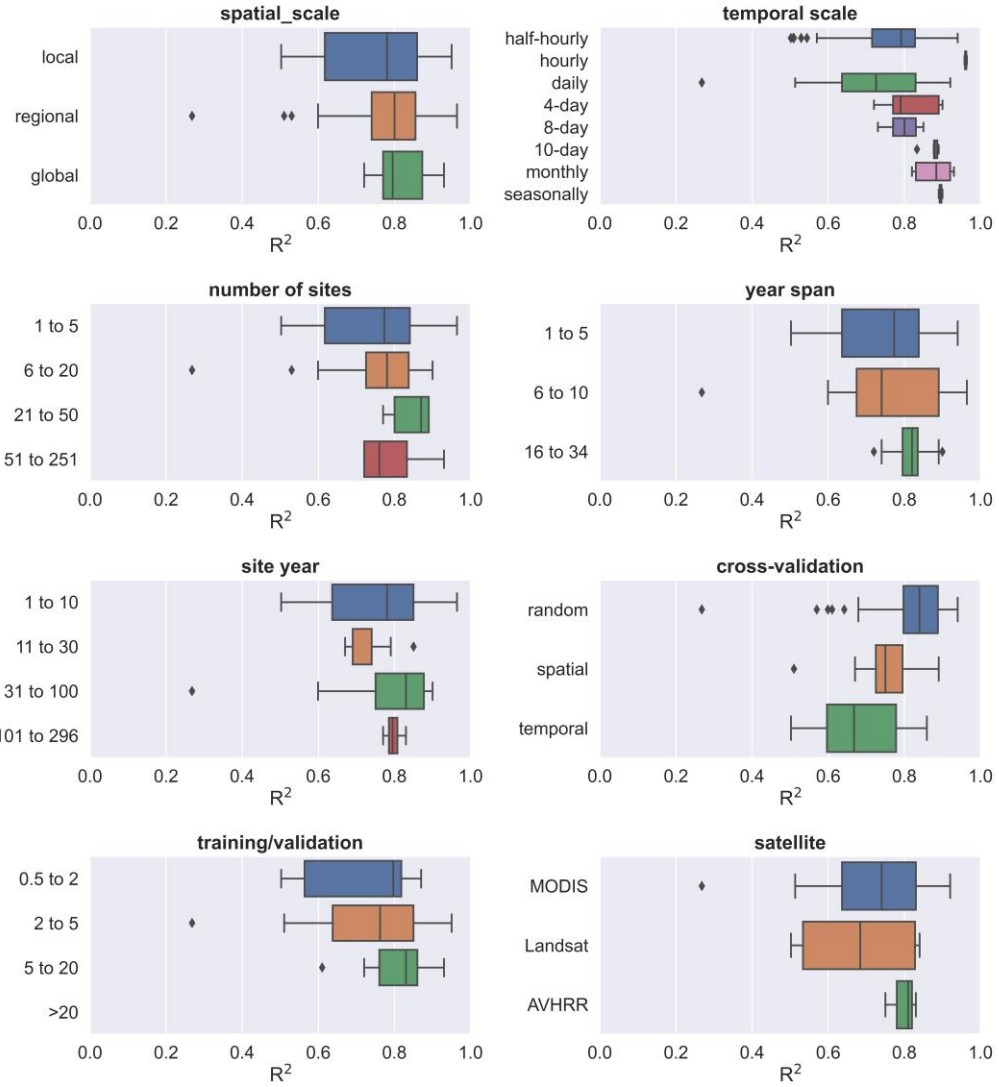

Figure 6. The effects of other model features (i.e. spatial scale, number of sites, temporal scale, year span, site year, validation method, training/validation ratio, and satellite imagery used) on the R-squared.

### 3.3.5 Linear correlation of quantitative features and R-squared

We also analyzed the linear correlation (Fig. 7) between multiple quantitative features and the R-squared. We found that the magnitude of the linear correlation coefficients between the use of predictor combinations and the R-squared was higher than other features. The use of the predictor combination 'Ta and Rn/Rs' significantly improved the model accuracy. 'Temporal scale', 'time span', 'training/validation ratio', and 'number of sites'

showed weak positive correlations with R-squared (not significant, p-value > 0.1). The positive correlation
between 'temporal scale' and R-squared is higher among these features, although not significant. It should also
be paid more attention to in future studies. The feature 'training/validation ratio' and 'time span' are also
positively correlated (although not significantly) with the R-squared, suggesting the importance of the volume of
data in the training set in a data-driven machine learning model. Larger 'training/validation ratio' and 'time span'
may correspond to greater proportional coverage of the scenarios/conditions in the training set over the
validation set, and thus correspond to higher accuracy.

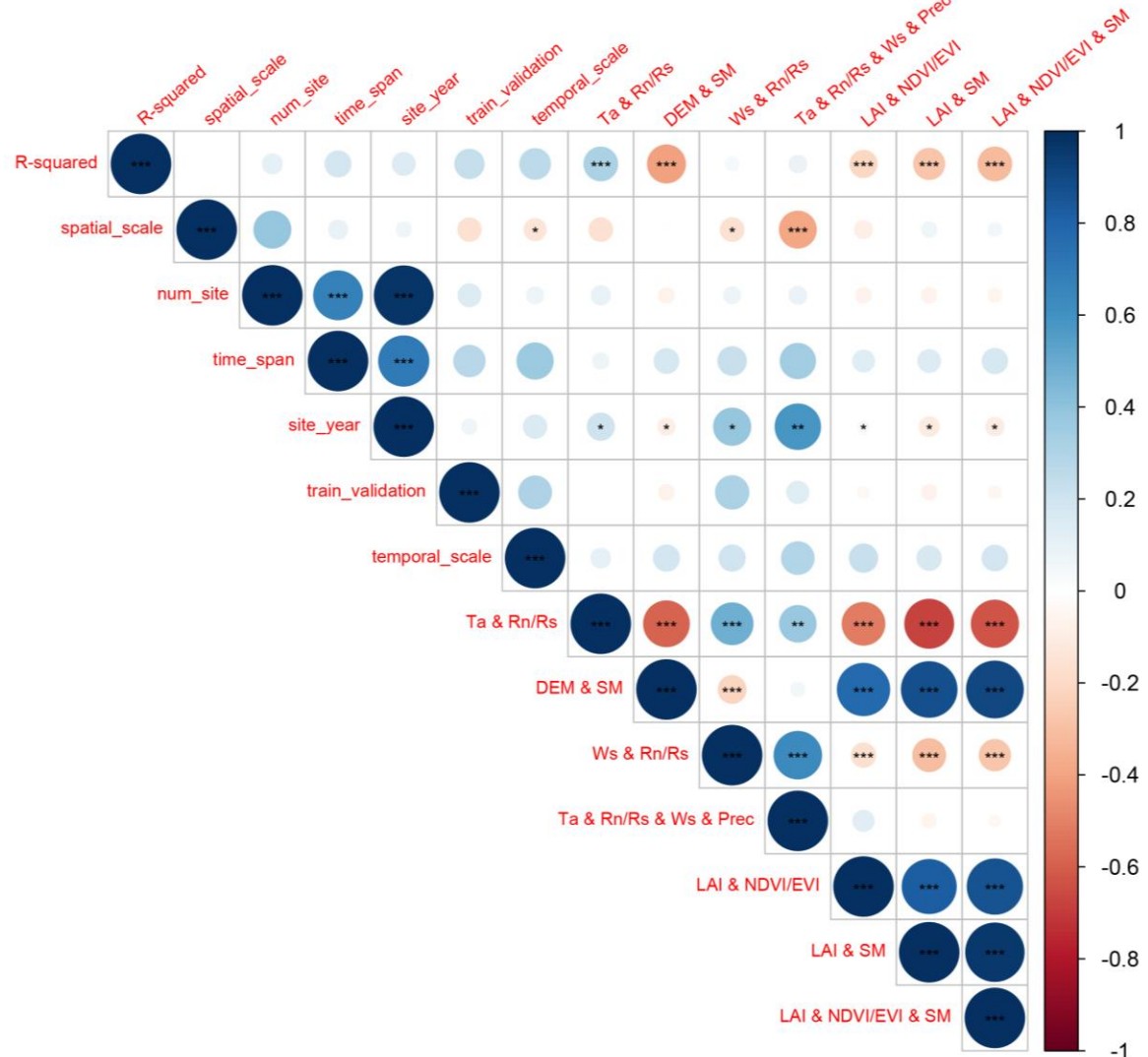


Figure 7. Evaluation of linear correlations between multiple features and the R-squared records with the
statistical significance test. For the feature 'spatial scale', the 'local' scale was set to 1, the 'regional' scale was
set to 2, and the 'global' scale was set to 3 in the analysis of linear correlation. For the use of various predictor
combinations with '&', the value for 'together used' is set as 1 and other conditions are set as 0 (e.g., for the
feature 'Ta & Rn/Rs & Ws & Prec', if Ta, Rn/Rs, Ws, and Prec were used together in the model, the value is set
as 1). Significance: the p-value < 0.01 (***), 0.05 (**), and 0.1 (*).

**4 Discussions**

With the accumulation of in situ EC observations around the world, the study of ET simulations based on data-driven approaches has received more attention from researchers in the last decade. Many studies have combined EC observations, various predictors, and machine learning algorithms to improve the prediction accuracy of water fluxes. To date, the results of these studies have not been comprehensively evaluated to provide clear guidance for feature selection in water flux prediction models. To better understand the approach and guide future research, we performed a meta-analysis of such studies. Machine learning-based water flux simulations and predictions still suffer from high uncertainty. By investigating the expected improvements that can be achieved by incorporating different features, we can avoid practices that may reduce model accuracy in future research.

**4.1 Opportunities and challenges in the water flux simulation**

In the above meta-analysis of the models, we found that water flux simulations based on EC observations can achieve high accuracy but also have high uncertainty through the modeling workflow. The R-squared of many water flux simulation models exceeds 0.8, possibly higher than some remote sensing-based and process-based models, and possibly higher than carbon flux simulations such as the net ecosystem exchange (NEE) in a similar modeling framework (Shi et al., 2022). This may be because many data on important variables affecting carbon flux such as soil and biomass pools, disturbances, ecosystem age, management activities, and land use history are not yet effectively and continuously measured (Jung et al., 2011) with the global spatially and temporally explicit information. While ET simulations rely on observations of moisture and energy conditions and vegetation conditions, much of the current available meteorological and remote sensing data have been effective to represent and capture the spatial and temporal dynamics of these predictors well.

**4.1.1 Comprehensive insights on model features**

Biophysical and meteorological variables are considered both important in ET simulations. This study found that models using a combination of meteorological variables had higher accuracy than models using only remotely sensed vegetation dynamic information. However, due to the high proportion of models with small temporal scales (e.g., half-hourly scale, hourly scale, and daily scale) in this study, this advantage of the combination of meteorological variables may be more suitable for small temporal scales. A possible explanation is that vegetation-related variables such as NDVI and LAI at the daily scale, 8-day scale, and 16-day scale have limited explanatory ability for hourly or daily-scale variability in ET, especially under cloudy conditions (e.g., tropical rainforest regions), the temporal continuity of the vegetation index data may be greatly limited (Zeng et al., 2022). This should be given more attention and some vegetation indices derived from hourly temporal resolution satellite remote sensing data such as GOES (Zeng et al., 2022) can be used for ET simulations to investigate the possible adding-values of vegetation indices at smaller time scales. In contrast, at a small temporal scale, the use of combinations of meteorological variables can capture moisture and energy conditions that control the rapid fluctuations of ET and thus has a dominant role in hourly or daily-scale ET prediction. This also corroborates the high accuracy of some physic-based ET estimation models (Rigden and Salvucci, 2015) that use only meteorological variables and not vegetation-related variables such NDVI (only an estimate

of vegetation height derived from land cover maps is used to represent vegetation conditions (Rigden and
Salvucci, 2015)).

There are differences in model accuracy among different PFTs. For example, in forest sites, limitations in data
accuracy of factors were possible because some remote sensing-based predictors such as NDVI, FAPAR, and
LAI have limited accuracy when applied to forest types (Liu et al., 2018b; Zeng et al., 2022). In addition, factors
such as crown density, which may significantly affect the proportion of soil evaporation, transpiration, and
evaporation of canopy interception, were not considered in these models, which may also lead to low model
accuracy. This suggests that in water flux simulation, the driving mechanisms of water fluxes in different PFTs
do affect the accuracy of machine learning models, and we need to consider more the actual and specific
influencing factors in specific PFTs. More variables that can quantify the ratio of evaporation and transpiration
should be considered for inclusion, which also appears to improve the mechanistic interpretability of such
machine learning models. A previous study (Zhao et al., 2019) combined the physics-based approach (e.g.,
Penman-Monteith equation) and machine learning to build hybrid models to improve interpretability. We should
make full use of empirical knowledge and experiences from process-based models to improve the accuracy and
interpretability of the machine learning approach.

Among the validation methods, random cross-validation has higher accuracy than spatial cross-validation and
temporal cross-validation. However, spatial cross-validation and temporal cross-validation may be able to better
help us recognize the robustness of the model when extrapolated (i.e., applied to new stations and new years).
The lower accuracy in the temporal cross-validation approach implies that we need to focus on interannual
hydrological and meteorological variability in the water flux simulations. In cropland sites, we may also need to
pay more attention to the effects of interannual variability in anthropogenic cropping patterns. If some extreme
weather years are not included, the robustness of the model when extrapolated to other years may be challenged,
especially in the context of the various extreme weather events of recent years. This can also inform the siting of
future flux stations. Regions where climate extremes may occur and biogeographic types not covered by
existing flux observation networks should be given more attention to achieve global-scale, accurate and robust
machine learning-based spatio-temporal prediction of water fluxes. Furthermore, although the R-squared and the
training/validation ratio show a positive correlation (Fig. 7) (i.e., a higher training/validation ratio may
correspond to a higher R-squared), we should still be cautious in reducing this ratio in our modeling. For a really
small validation set, it would be very challenging to determine which model is better given the potential
uncertainty caused by the considerable randomness.
**4.1.2 Differences from NEE predictions in the similar model framework**
In general, predictors related to meteorological, vegetation, and soil conditions were common to both ET and
NEE simulations in a similar framework (Shi et al., 2022). However, in NEE predictions, explanatory variables
such as soil organic content, photosynthetic photon flux density, and growing degree days (Shi et al., 2022) are
not necessary for ET predictions. The selection of these variables requires our prior knowledge of the dominant
drivers of ET and NEE anomalies of particular ecosystems and their differences.

The accuracy of NEE predictions (Shi et al., 2022) can be more limited by global variability across biomes and
locations (Nemani et al., 2003) given the lack of locally measured data on soil and biomass pools, disturbances,
ecosystem age, management activities and land use history (Jung et al., 2011). It can result in a higher
heterogeneity of the training data in large-scale modeling with multiple flux sites (Shi et al., 2022) and the weak
ability to capture the NEE anomalies. In contrast, in ET predictions, meteorological variables and vegetation
conditions appear to be already sufficient to capture a considerably large fraction of the ET variations in most
conditions.

In future ET prediction studies, given that few current ET products have time scales smaller than daily scale
(Jung et al., 2019; Pan et al., 2020), improvements in the accuracy of daily and hourly models may be necessary
to fill this gap. Besides, the partitioning of ET components (i.e., transpiration, interception evaporation, and soil
evaporation) can be more focused to better decouple the contributions of vegetation and soil to ET with machine
learning (Eichelmann et al., 2022). It can be further matched with the partitioning of NEE (i.e., to GPP and
ecosystem respiration) to increase our knowledge of the global water cycle and ecosystem functioning and
obtain further refined global carbon-water fluxes coupling relations (Eichelmann et al., 2022). Also, the above
two promising improvements can be beneficial for research on topics related to the global terrestrial water cycle
(Fisher et al., 2017).
**4.2 Uncertainties and limitations of this meta-analysis**
**4.2.1 The limited number of available literature and model records**
Despite many articles and model records collected through our efforts to perform this meta-analysis, there still
appears to be a long way to go to finally and completely understand the various mechanisms involved in water
flux simulation with machine learning. Some of the insights provided by this study can be not robust (due to the
limited sample size available when the goal is to assess the effects of multiple features), but this does not negate
the fact that this study does obtain some meaningful findings. Therefore, researchers should treat the results of
this study with caution, as they were obtained only statistically. Overall, it is still positive to conduct a meta-
analysis of such studies, considering their rapid growth in number and lack of guiding directions.
**4.2.2 Publication bias and weighting**
Publication bias and weighting: Due to the relatively limited number of articles that could be included in the
meta-analysis, this study did not focus much on publication bias. Meta-analytic studies in other fields typically
measure the quality of journals and the public availability of research data (Borenstein et al., 2011; Field and
Gillett, 2010) to determine the weighting of the literature in a comprehensive assessment. However, most of the
articles did not publicly provide flux observations or share developed models. Meta-analysis studies in other
fields typically measure the impact of included studies based on sample size and variance of experimental
results (Adams et al., 1997; Don et al., 2011; Liu et al., 2018a). In this study, due to the lack of a convincing
manner to determine weights among articles, we assigned the same weight to the results for all the literature.
**4.2.3 Uncertainties in the information of the extracted features**
At the information extraction level, the following issues may also introduce uncertainties:
a)  Uncertainties caused by data quality control (e.g. gap-filling (Hui et al., 2004)) are difficult to assess
effectively. Gap-filling is a commonly used technique to fill in low-quality data in flux observations.
However, the impact of this practice on machine learning-based ET prediction models is unclear, due to the
difficulty of directly assessing how this technique is performed in various studies by this meta-analysis.
Typically, models with small time scales (e.g., hourly scale, daily scale) can exclude low-quality
observations and use only high-quality data. However, for models with large time scales (e.g., monthly
scales), gap-filling (e.g., based on meteorological data) may be unavoidable. This may lead to a decrease in
training data purity and introduce uncertainty in the subsequent prediction model development.
b)  Systematic uncertainties caused by the energy balance closure (EBC) issue in eddy-covariance flux
measurements are also difficult to assess by this meta-analysis. EBC is a common problem (Eshonkulov et
al., 2019) in eddy-covariance flux observations. For that reason, the latent heat flux measured potentially
underestimates ET. Some prediction models corrected EBC (e.g., using Bowen ratio preserving (Mauder et
al., 2013, 2018) and energy balance residuals (Charuchittipan et al., 2014; Mauder et al., 2018)) in the
processing of training data, but some did not. How this will affect the accuracy of the prediction model is
not clear due to multiple factors that need to be evaluated that influence EBC (Foken, 2008), including
measurement errors of the energy balance components, incorrect sensor configurations, influences of
heterogeneous canopy height, unconsidered energy storage terms in the soil-plant-atmosphere system,
inadequate time averaging intervals, and long-wave eddies (Jacobs et al., 2008; Foken, 2008; Eshonkulov
et al., 2019). To reduce this uncertainty, more attention to flux site characteristics (Eshonkulov et al., 2019)
related to PFT, topography, flux footprint area, etc., to select the appropriate correction method is
necessary for future studies.
c)  As most studies used far more water flux observation records than the number of covariates in their
regression models, we did not adjust the R-squared in this study to an adjusted R-squared.
d)  The various specific ways in which the parameters of the model are optimized are not differentiated. They
are broadly categorized into different families or kinds of algorithms, which may also introduce uncertainty
into the assessment.
e)  The assessment of some features is not detailed due to the limitations of the available model records. For
example, the classification of PFT could be more detailed. 'Forest' could be further classified as broadleaf
forest, coniferous forest, etc. while 'cropland' could be further classified as rainfed and irrigated cropland
based on differences in their response mechanisms of water fluxes to environmental factors.

## 5 Conclusion

We performed a meta-analysis of the water flux simulations combining in situ flux observations from flux
stations/networks, meteorological, biophysical, and ancillary predictors, and machine learning. The main
conclusions are as follows:
1.  SVM (average R-squared = 0.82) and RF (average R-squared = 0.81) outperformed over evaluated
algorithms with sufficient sample size in both cross-study and intra-study (with the same training dataset)
comparisons.
2.  The average accuracy of the model applied to arid regions is higher than in other climate types.
3.  The average accuracy of the model was slightly lower for forest sites (average R-squared = 0.76) than for
cropland and grassland sites (average R-squared = 0.8 and 0.79), but higher than for shrub sites (average R-
squared = 0.67).
4.  Among various predictor variables, the use of Rn/Rs, Prec, Ta, and FAPAR improved the model accuracy.
The combination of Ta and Rn/Rs is very effective especially in the forest type, while in the grassland type
the combination of Ws and Rn/Rs is also effective.
5.  Among the different validation methods, random cross-validation shows higher model accuracy than spatial
cross-validation and temporal cross-validation.

**Acknowledgements**
We thank the editor and two anonymous reviewers for their insightful comments which contributed substantially
to the improvement of this manuscript.
**Financial support**
This research was supported by the National Natural Science Foundation of China (Grant No. U1803243), the
Key projects of the Natural Science Foundation of Xinjiang Autonomous Region (Grant No. 2022D01D01), the
Strategic Priority Research Program of the Chinese Academy of Sciences (Grant No. XDA20060302), and High-
End Foreign Experts Project.
**Author Contributions**
HS and GL were responsible for the conceptualization, methodology, formal analysis, investigation, visualization,
and writing. OH contributed to the investigation. XM, XY, YW, WZ, MX, CZ and YZ processed the data. AK,
TVDV and PDM provided supervision.
**Competing interests**
The authors declare that they have no conflict of interest.
**Code availability**
The codes that were used for all analyses are available from the first author (shihaiyang16@mails.ucas.ac.cn)
upon request.
**Data availability**
The data used in this study can be accessed by contacting the first author (shihaiyang16@mails.ucas.ac.cn) upon
request.

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
