# Peer review of "Evaluation of water flux predictive models developed using eddy"

_Hydrology and Earth System Sciences, 2022_

## Author Response (AR1)

**Response to Referee #1**

The authors conducted a meta-analysis to evaluate the performance of machine learning (ML) algorithms in the estimation of evapotranspiration. I believe this topic is timely and of interest to the HESS community. The motivation of the study, method, and results are clearly outlined, and they reach clear conclusions. Overall, this manuscript is informative and well structured. However, I believe there are several minor aspects which can be improved. Therefore, I support its publication in HESS with minor revisions.

Response: We would like to thank the reviewer for the positive comments and the time invested to review our manuscript. The revised manuscript will follow the reviewer's recommendations.

1) L34 "ET is the most important indicator of the water cycle": ET is not an indicator. It is a water balance component. Also, it may be not the most important component. I suggest writing "ET is one of the most important components of the water cycle ~"

Response: Thank you for the insightful comments. It will be revised as 'ET is one of the most important components of the water cycle'.

Action: revised as 'Evapotranspiration (ET) is one of the most important components of the water cycle in terrestrial ecosystems.'

2) L51-53: add examples and references to support the argument.

Response: Two references will be added: 'For remote sensing-based physical models and process-based land surface models, some physical processes have not been well characterized due to the lack of understanding of the detailed mechanisms influencing ET under different environmental conditions. For example, the inaccurate representation and estimation of stomatal conductance (Li et al., 2019) and the linearization (McColl, 2020) of the Clausius-Clapeyron relation in the Penman-Monteith equation may introduce both empirical and conceptual errors into estimates of ET.'

Action: elaborated as 'For example, the inaccurate representation and estimation of stomatal conductance (Li et al., 2019) and the linearization (McColl, 2020) of the Clausius-Clapeyron relation in the Penman-Monteith equation may introduce both empirical and conceptual errors into estimates of ET.'

3) L82: define NDVI, EVI and LAI.

Response: It will be defined as Normalized Difference Vegetation Index (NDVI), Enhanced Vegetation Index (EVI), and Leaf area index (LAI).

Action: Defined.

4) L83: define GPP

Response: It will be defined as 'Gross Primary Productivity.

Action: Defined.

5) L153-155: I agree with the authors' point, but RMSE is still an important measure of the model performance. I think there is a way to normalize the RMSE when the magnitude or

standard deviation of water flux are available. If possible, I recommend analyzing RMSE as well.

Response: Thank you for the insightful comments. The RMSE depends on the magnitude of the ET value of the training data. For example, due to the difference in the range of ET values, models developed from flux stations in dry grasslands will typically have lower RMSE than models developed by flux stations based on forests in humid regions. Therefore, RMSE may not be a good metric for cross-study comparisons. We will clarify this issue in the revised manuscript. Since we do not have the raw data of these studies, it is difficult to unify the differences in RMSE across data sets in a normalized way.

Mean Absolute Percentage Error (MAPE) can be useful but not commonly used or reported as R-squared in such studies.

Action: clarified as 'Although RMSE is also often used for model accuracy assessment, its dependence on the magnitude of water flux values makes it difficult to use for fair comparisons between studies. For example, due to the difference in the range of ET values, models developed from flux stations in dry grasslands will typically have lower RMSE than models developed by flux stations based on forests in humid regions. Therefore, RMSE may not be a good metric for cross-study comparisons.'

6) L225-229 and Figure 5 and Figure7: I think the authors should discuss variables which decrease the performance of the ML models (NDVI etc.). To do this, the authors may need to refer to Figure 7. Therefore, I suggest reordering Figures (i.e., 7 ->6 and 6->7). Figure 7 implies performance decreases due to NDVI (and other variables) may be spurious. In order to overcome such limitations, I suggest performing additional analysis by grouping ML models which use Rn/Rs and Ta and then generating Figure 5.

Response: Thank you for the insightful comments. This is a good suggestion. We will consider adjusting the order of the figures based on your comments and will perform additional analysis by grouping ML models which use Rn/Rs and Ta as you suggested.

Action: We placed Figure 7 in the supplementary material as Figure S2 and replaced Figure 5 with an figure of evaluation of the combination of predictor variables. Figure 5 was placed in the additional material as Figure S1. In the revised manuscript, we assessed the impact of predictor variables on model accuracy at two levels: (1) the correspondence between the use or non-use of individual predictor variables and model accuracy, and (2) after analyzing the dependence of the use of predictor variables, we analyze the impact of combinations of predictor variables (distinguishing mainly between meteorological predictor combinations and remote sensing-based vegetation-related predictor combinations).

This paragraph was elaborated as:

[revised manuscript text omitted]

7) Figure5: difficult to compare variables. I think visualization can be improved by grouping variables which improve performance or not.

Response: We will consider adjusting the order of the figures based on your comments, and will perform additional analysis by grouping variables as you suggested (also based on findings in Fig. 7).

Action: In the revised manuscript, we evaluated the effect of the combination of predictor variables (please refer to the above response/action in the last comment).

8) L261-263: I cannot agree. Data-driven approach and process-based approach are complementary. This should be revised.

Response: We will modify the description here. Indeed data-driven and process-based approaches are complementary and both are rapidly developing and therefore of equal importance in the future direction of ET estimation.

Action: revised as 'With the accumulation of in situ EC observations around the world, the study of ET simulations based on data-driven approaches has received more attention from researchers in the last decade. Many studies have combined EC observations, various predictors, and machine learning algorithms to improve the prediction accuracy of site-scale water fluxes.'

9) L336-338: As the authors briefly mentioned here, eddy covariance observations are subject to random, gap-filling, and systematic (energy balance closure) uncertainty. There are several ways to address this uncertainty. For example, some studies may use a gap-filled dataset but some studies may choose observation only. Also, the energy balance closure problem can be addressed differently (uncorrected, Bowen-ratio corrected, and use of energy balance residual). Depending on this choice, the performance of ML algorithms may vary significantly (particularly energy closure problem is important). Although the authors mentioned observational uncertainty as a limitation of this research in L336-338, I believe this brief mention is not enough. If you can extract this information from the literature, I suggest performing an additional analysis (e.g., performance comparison for energy balance corrected vs uncorrected). If it is indeed difficult to extract the information from the literature, this topic should be discussed more thoroughly at least.

Response: We will elaborate on the discussion section on this issue. Indeed uncertainties in the observations (including those in Gap-filling) may affect model accuracy. The energy closure problem does also confuse researchers in this field which may lead to the underestimation of ET values, although some datasets (e.g., FLUXNET) have provided observations of latent heat after bias correction in energy closure.

When the problem of energy closure is not negligible, the use of energy balance uncorrected data may affect the model accuracy. We will discuss this issue further based

on previous studies (combined with the potential severity of the bias in ET observations caused by the energy closure problem in various environmental conditions).

Action: the discussion section 4.2.3 was elaborated:

(a) Uncertainties caused by data quality control (e.g. gap-filling (Hui et al., 2004)) are difficult to assess effectively. Gap-filling is a commonly used technique to fill in low-quality data in flux observations (Chen et al., 2012; Hui et al., 2004). However, the impact of this practice on machine learning-based ET prediction models is unclear, due to the difficulty of directly assessing how this technique is performed in various studies by this meta-analysis. Typically, models with small temporal scales (e.g., hourly scale, daily scale) can exclude low-quality observations and use only high-quality data. However, for models with large time scales (e.g., monthly scales), gap-filling (e.g., based on meteorological data) may be unavoidable. This may lead to decrease in training data purity and introduce uncertainty in the subsequent prediction model development.

(b) Systematic uncertainties caused by the energy balance closure (EBC) issue in eddy-covariance flux measurements are also difficult to assess by this meta-analysis. EBC is a common problem (Eshonkulov et al., 2019) in eddy-covariance flux observations. For that reason, the latent heat flux measured potentially underestimates ET. Some prediction models corrected EBC (e.g., using Bowen ratio preserving (Mauder et al., 2013, 2018) and energy balance residuals (Charuchittipan et al., 2014; Mauder et al., 2018)) in the processing of training data, but some did not. How this will affect the accuracy of the prediction model is not clear due to multiple factors that need to be evaluated that influence EBC (Foken, 2008), including measurement errors of the energy balance components, incorrect sensor configurations, influences of heterogeneous canopy height, unconsidered energy storage terms in the soil-plant-atmosphere system, inadequate time averaging intervals, and long-wave eddies (Jacobs et al., 2008; Foken, 2008). To reduce this uncertainty, more attention to flux site characteristics (Eshonkulov et al., 2019) related to PFT, topography, flux footprint area, etc., to select the appropriate correction method is necessary for future studies.

**Response to Referee #2**

In this study, Shi et al., presented a meta-analysis of the performance of machine learning (ML) algorithms in the estimation of evapotranspiration. While this manuscript is interesting and within the scope of HESS, I have a few major concerns.

Response: We would like to thank the reviewer for the positive comments and the time invested to review our manuscript. The revised manuscript will follow the reviewer's recommendations.

Most importantly, while this is a meta-analysis, the authors were comparing results from different publications, in which different data sets and sites may have been used. That being said, some of the results are not directly comparable. For example, Zeng et al. 2020, may have selected a few sites that are much more difficult to predict; and can not be compared with the results presented in another publication. Also, some sites may use in-situ estimates of LAI and VIs, while others use LANDSAT or even MODIS LAI and VIs. In order to make their results publishable, they need to find a way to harmonize the data sets used in all studies. Or, they need to justify that they have an inclusion criteria when selecting all publications (instead of just stating we searched on Scopus). In addition, I am not sure whether the number of models they chose can well support their comparison of so many features.

Response: Thank you for the insightful comments. Some studies have indeed used sites that are difficult to predict. Usually, with meta-analysis, we only get comprehensive findings, and it is difficult to improve the understanding of extreme and exceptional cases (because the mean or median of statistical results is what we used in the formal assessment). The inclusion of extreme cases (such as the very unpredictable sites you mentioned) may negatively affect the evaluation results, but this negative effect may be limited if they only share a low proportion of the samples.

In addition, there are comparisons of studies using the same data (but different algorithms) (Fig. 3b) in this study. The difference in the data between studies is constrained (keeping other features the same but only the algorithms different): Fig 3a included various conditions across studies, i.e., what the reviewers raised; Fig 3b is the result of a comparison of model cases based on the same data and different machine learning algorithms, and is a correction and a more objective characterization of the issue with Fig 3a.

Few studies have used in-situ measured LAIs and VIs for modeling, as this is not helpful for the large-scale, long-time series predictions compared to remote sensing-based LAIs and VIs. Regarding these worries, we will clarify the details of these inclusion criteria which were used for the screening of the article in the revised manuscript.

Although multiple features were evaluated in this study, there are only a few features that predominantly affect the accuracy of the model. Some features may be insignificant (only weakly influencing) and we will consider deleting these features to highlight the analysis of

the major influencing features. In addition, we have included as large a sample as possible to support our findings, and our findings for the meta-analysis of ET predictions are likely to be more robust as further such studies are added in the future.

**Actions:**

**Action 1:**
Further response:

The purpose of meta-analysis is to combine the heterogeneity of studies to obtain comprehensive findings. If we filter all articles to use the criteria 'using the same data and sites', then few articles can be included in the meta-analysis and this analysis will be difficult to implement.

The paper by Zeng et al. uses global-scale data from FLUXNET (which simulates carbon fluxes rather than water flux) with high variations in site conditions, some sites of which may indeed be much more difficult to predict. Such a global scale using FLUXNET data is the largest in such studies and belongs to the outlier/extreme cases. This meta-analysis gave a general/average reference for such studies, and for very large scale, or otherwise specific studies, researchers should still focus on the specificity of their models.

On the feasibility of the methodology to assess the impacts of various model features on model accuracy by meta-analysis, there are several published articles (listed below) using a similar methodology in the cross-study comparisons (comparison of models developed in various studies despite the different data and features used).

- Khatami, R., Mountrakis, G., and Stehman, S. V.: A meta-analysis of remote sensing research on supervised pixel-based land-cover image classification processes: General guidelines for practitioners and future research, Remote Sensing of Environment, 177, 89–100, https://doi.org/10.1016/j.rse.2016.02.028, 2016.
- Zolkos, S. G., Goetz, S. J., and Dubayah, R.: A meta-analysis of terrestrial aboveground biomass estimation using lidar remote sensing, Remote Sensing of Environment, 128, 289–298, https://doi.org/10.1016/j.rse.2012.10.017, 2013.

Therefore, if the objective is to obtain a comprehensive understanding and general guidelines, we think the meta-analysis methodology of this study can be feasible (although some special cases should also be attended to).

We hope you can agree on the value of this meta-analysis, although some of such studies can vary widely and thus were not very comparable.

**Action 2:**
We clarified the detailed inclusion criteria when selecting all publications and the way to harmonize the data sets used in all studies:

Revised as:

[revised manuscript text omitted]

Also, the authors have another paper looking at similar topics (even with some similar

pictures and texts) in discussion on Biogeoscience. As an example, in this paper:

Line 114-117: And in machine learning, in general, modeling with unbalanced data (with significant differences in the distribution between the training validation sets) may result in lower model accuracy.

And in the BG paper:

Line 91-94: Modeling with unbalanced data (where the difference between the distribution of the training and validation sets is significant even if selected at random) may result in lower model accuracy.

The only differences between the two papers is that the BG paper focused on NEE, while this paper looked into ET. I am not sure whether it is acceptable to publish two somewhat similar papers in two different EGU journals.

Response: Thank you for the insightful comments. Although the framework/methods of these two manuscripts are similar, the topics are different. One is NEE (which is a carbon cycle-related topic) and the other is ET (which is a water cycle-related topic). NEE and ET are not directly correlated and therefore need to be studied separately. The NEE and ET prediction use different explanatory variables and analysis/discussions of their mechanisms are also different. The potential readers of these two manuscripts are also different. We will carefully check for possible duplicate text.

Action: We have checked for possible duplicate text in these two manuscripts.

At the same time, overall, the writing of the manuscript is good. But I do find it difficult to follow from time to time. For example, the authors used many abbreviations without defining them (EVI, GPP and NDVI), and some of them may not be very familiar with all the readers.

Response: Abbreviations will be defined such as Normalized Difference Vegetation Index (NDVI), Enhanced Vegetation Index (EVI), and Leaf area index (LAI).

Action: These abbreviations were defined.

Minor comments:

Line 34: I suggest that the authors refrain from statements like this, precipitation and runoff are at least equally important;

Response: Thank you for the insightful comments. It will be corrected as 'ET is one of the most important components of the water cycle in terrestrial ecosystems.'

Action: Corrected as 'Evapotranspiration (ET) is one of the most important components of the water cycle in terrestrial ecosystems.'

Line 52: detailed?

Response: Two references will be added (for the detailed limitations in physics-based methods): 'For remote sensing-based physical models and process-based land surface models, some physical processes have not been well characterized due to the lack of understanding of the detailed mechanisms influencing ET under different environmental conditions. For example, the inaccurate representation and estimation of stomatal

conductance (Li et al., 2019) and the linearization (McColl, 2020) of the Clausius-Clapeyron relation in the Penman-Monteith equation may introduce both empirical and conceptual errors into estimates of ET.'

Action: elaborated as 'For example, the inaccurate representation and estimation of stomatal conductance (Li et al., 2019) and the linearization (McColl, 2020) of the Clausius-Clapeyron relation in the Penman-Monteith equation may introduce both empirical and conceptual errors into estimates of ET.'

Line 155: I still do not understand why RMSEs are not used.
Response: The RMSE depends on the magnitude of the ET value of the training data. For example, due to the difference in the range of ET values, models developed from flux stations in dry grasslands will typically have lower RMSE than models developed by flux stations based on forests in wet areas. Therefore, RMSE may not be a good metric for cross-study comparisons. We will clarify this issue in the revised manuscript.
Action: clarified as 'Although RMSE is also often used for model accuracy assessment, its dependence on the magnitude of water flux values makes it difficult to use for fair comparisons between studies. For example, due to the difference in the range of ET values, models developed from flux stations in dry grasslands will typically have lower RMSE than models developed by flux stations based on forests in humid regions. Therefore, RMSE may not be a good metric for cross-study comparisons.'

L186: outperformed whom? I believe that similar issues can be found in other places of the manuscript.
Response: The 'outperform' here refers to the higher accuracy of SVM and RF compared to other algorithms in Fig. 3a. We will further check other such descriptions in this manuscript.
Action: revised as 'SVM and RF outperformed (Fig. 3a) across studies (better than other algorithms with sufficient sample size in Fig. 3a such as ANN).'

'With sufficient sample size' were also added to other such descriptions.

---

## Author Response (AR2)

**Response to Referee #2** (hess-2022-90)

In the revised version of this manuscript, the authors have addressed most of my concerns raised in my first review. Overall, I still believe that the authors need to make a clear distinction between this paper and the BG paper. I have a few suggestions, before this manuscript could be further considered for publication, please accept them as constructive criticism.

Response: Thank you very much for your insightful comments, and we have revised this manuscript based on your concerns.

1, as a reader, I may ask: why can these two papers be just one paper? These two meta-analyses are based on nearly identical sampling of studies. As a way to address this, the authors could try to add a few papers that only models ET;

Response & Actions: We performed these two meta-analyses separately because the mechanisms of the NEE and ET anomalies, and the predictor variables used, can be considerably different. Only one paper (Jung et al., 2011) modeled both ET and Net ecosystem exchange (NEE) was included in both these two meta-analyses, and the overlap between the papers included in the two meta-analyses was very low. The vast majority of papers in this meta-analysis modeled only ET.

| Meta-analysis | Papers included |
|---|---|
| BG (40 NEE papers) | (Berryman et al., 2018; Braybrook et al., 2021; Cho et al., 2021; Cleverly et al., 2020; Cui et al., 2021; Evrendilek, 2013; Fu et al., 2014, 2009; Huemmrich et al., 2019; Ichii et al., 2017; **Jung et al., 2011**; Kato and Tang, 2008; Kondo et al., 2015; Krasnova et al., 2019; Liu et al., 2016, 2018; Lucas-Moffat et al., 2018; Madani et al., 2017; Melesse and Hanley, 2005; Moffat et al., 2010; Mueller et al., 2010; Papale and Valentini, 2003; Park et al., 2018; Reed et al., 2021; Reitz et al., 2021; Ryu et al., 2018; Schubert et al., 2010; Stiegler et al., 2019; Sun et al., 2020, 2019; Teklemariam et al., 2010; Tian et al., 2017; Tramontana et al., 2016; Ueyama et al., 2013; Virkkala et al., 2021; Xiao et al., 2008; Zeng et al., 2020; Zhang et al., 2014; Zhou et al., 2020) |
| HESS (32 ET papers) | (Bai et al., 2021; Dou and Yang, 2018, 2017; Fang et al., 2020; Feng et al., 2020; Gerken et al., 2019; Granata, 2019; Granata and Di Nunno, 2021; Guo et al., 2019; **Jung et al., 2011**; Kafer et al., 2020; Li et al., 2018, 2021; Lu and Zhuang, 2010; Pang et al., 2021; Papale et al., 2015; Qin et al., 2005b, a; Safa et al., 2018; Shang et al., 2021; Van Wijk and Bouten, 1999; Vrugt et al., 2002; Vulova et al., 2021; Wang et al., 2021a, b; Xie et al., 2021; Xu et al., 2018; Yin et al., 2021; Zhang et al., 2021, 2020; Zhao et al., 2019) |

2, at the same time, I believe the users can open up by addressing the differences between the predictions of these two fluxes in the discussion. What are the major differences between the predictions, is it generally easier to predict ET?

Response & Actions: This is a good suggestion. Here we simply discussed the major differences

between the predictions in the beginning discussion:

'In the above meta-analysis of the models, we found that water flux simulations based on EC observations can achieve high accuracy but also have high uncertainty through the modeling workflow. The R-squared of many water flux simulation models exceeds 0.8, possibly higher than some remote sensing-based and process-based models, and possibly higher than carbon flux simulations such as the net ecosystem exchange (NEE) in a similar modeling framework (Shi et al., 2022). This may be because many data on important variables affecting carbon flux such as soil and biomass pools, disturbances, ecosystem age, management activities, and land use history are not yet effectively and continuously measured (Jung et al., 2011) with the global spatially and temporally explicit information. While ET simulations rely on observations of moisture and energy conditions and vegetation conditions, much of the current available meteorological and remote sensing data have been effective to represent and capture the spatial and temporal dynamics of these predictors well.' (line 325-334)

Also, the discussion of major differences between the predictions is added as the additional section 4.1.2:

**4.1.2 Differences from NEE predictions in the similar model framework**

'In general, predictors related to meteorological, vegetation, and soil conditions were common to both ET and NEE simulations in a similar framework (Shi et al., 2022). However, in NEE predictions, explanatory variables such as soil organic content, photosynthetic photon flux density, and growing degree days (Shi et al., 2022) are not necessary for ET predictions. The selection of these variables requires our prior knowledge of the dominant drivers of ET and NEE anomalies of particular ecosystems and their differences.

The accuracy of NEE predictions (Shi et al., 2022) can be more limited by global variability across biomes and locations (Nemani et al., 2003) given the lack of locally measured data on soil and biomass pools, disturbances, ecosystem age, management activities and land use history (Jung et al., 2011). It can result in a higher heterogeneity of the training data in large-scale modeling with multiple flux sites (Shi et al., 2022) and the weak ability to capture the NEE anomalies. In contrast, in ET predictions, meteorological variables and vegetation conditions appear to be already sufficient to capture a considerably large fraction of the ET variations in most conditions.

In future ET prediction studies, given that few current ET products have time scales smaller than daily scale (Jung et al., 2019; Pan et al., 2020), improvements in the accuracy of daily and hourly models may be necessary to fill this gap. Besides, the partitioning of ET components (i.e., transpiration, interception evaporation, and soil evaporation) can be more focused to better decouple the contributions of vegetation and soil to ET with machine learning (Eichelmann et al., 2022). It can be further matched with the partitioning of NEE (i.e., to GPP and ecosystem respiration) to increase our knowledge of the global water cycle and ecosystem functioning and obtain further refined global carbon-water fluxes coupling relations (Eichelmann et al., 2022). Also, the above two promising improvements can be beneficial for research on topics related to the global terrestrial water cycle (Fisher et al., 2017).' (line 383-406)

3, In the current version, the authors added a section of "Linear correlation of quantitative features

and R-squared". Some of the arguments in this section are somewhat controversial to me. For instance, the authors seem to be arguing that a higher ratio towards the training set would lead to a higher $R^2$. However, when there's a really small validation set, it would be very challenging to determine which model is better (very random). I.e. the authors of these studies may actually look only at the testing set and it might not be a good practice.

Response & Actions: It is elaborated in the discussion section:

'Among the validation methods, random cross-validation has higher accuracy than spatial cross-validation and temporal cross-validation. However, spatial cross-validation and temporal cross-validation may be able to better help us recognize the robustness of the model when extrapolated (i.e., applied to new stations and new years). The lower accuracy in the temporal cross-validation approach implies that we need to focus on interannual hydrological and meteorological variability in the water flux simulations. In cropland sites, we may also need to pay more attention to the effects of interannual variability in anthropogenic cropping patterns. If some extreme weather years are not included, the robustness of the model when extrapolated to other years may be challenged, especially in the context of the various extreme weather events of recent years. This can also inform the siting of future flux stations. Regions where climate extremes may occur and biogeographic types not covered by existing flux observation networks should be given more attention to achieve global-scale, accurate and robust machine learning-based spatio-temporal prediction of water fluxes. Furthermore, although the R-squared and the training/validation ratio show a positive correlation (Fig. 7) (i.e., a higher training/validation ratio may correspond to a higher R-squared), we should still be cautious in reducing this ratio in our modeling. For a really small validation set, it would be very challenging to determine which model is better given the potential uncertainty caused by the considerable randomness.' (line 368-382)

Monir comments:

Figure 3: $R^2$

Response & Actions: modified (and also modified in other figures)

L340-341: it should be paid more attention to?

Response & Actions: elaborated as:

'A possible explanation is that vegetation-related variables such as NDVI and LAI at the daily scale, 8-day scale, and 16-day scale have limited explanatory ability for hourly or daily-scale variability in ET, especially under cloudy conditions (e.g., tropical rainforest regions), the temporal continuity of the vegetation index data may be greatly limited (Zeng et al., 2022). This should be given more attention and some vegetation indices derived from hourly temporal resolution satellite remote sensing data such as GOES (Zeng et al., 2022) can be used for ET simulations to investigate the possible adding-values of vegetation indices at smaller time scales.' (line 340-346)

L395: remove the ();

Response & Actions: removed

L425: I do not know whether all the site PIs care about machine learning at global scale. I think a better suggestion may be for the networks;

Response & Actions:

revised as you suggested: 'We performed a meta-analysis of the water flux simulations combining in situ flux observations from flux stations/networks, meteorological, biophysical, and ancillary predictors, and machine learning.' (line 458)

---

## Author Response (AR3)

**Reviewer comments:**

In the revised manuscript, I believe that most of my concerns from my previous rounds have been addressed. I have no further comments and do appreciate the authors' efforts to improve this manuscript. As a reminder, please make sure to check so that there are no similar sentences in the two papers.

**Response & Actions:**

Thank you for your suggestion. By using the 'compare' function of MS Word, we have checked (on the next attached page) and modified the found similar/duplicate sentences. The following changes were made to similar sentences to make the text different enough from the BG paper.

Similar sentences/ duplicate words (red)
Revised (blue)

[revised manuscript text omitted]

**1 Introduction**

Net ecosystem exchange (NEE) of $CO_2$ is an important indicator of carbon cycling in terrestrial ecosystems (Fu et al., 2019), and accurate estimation of NEE is important for the development of global carbon neutral policies. Although process-based models have been used for NEE simulations (Mitchell et al., 2009), their accuracy and spatial resolutions of the model outputs are limited probably due to the lack of understanding and quantification of complex processes. Many researchers have tried to use a data-driven approach as an alternative (Fu et al., 2014; Tian et al., 2017; Tramontana et al., 2016; Jung et al., 2011). On the one hand, it was made possible by the increase in the growth of global carbon flux observations and the large amount of flux observation data being accumulated. Since the 1990s, the use of the eddy covariance technique to monitor NEE has been rapidly promoted (Baldocchi, 2003). Several regional and global flux measurement networks have been established for the big data management of the flux sites, including CarboEuro flux (Europe), AmeriFlux (North America), OzFlux (Australia), ChinaFlux (China), FLUXNET (global), etc. On the other hand, machine learning approaches are increasingly used to extract patterns and insights from the ever increasing stream of geospatial data (Reichstein et al., 2019). The rapid development of various algorithms and high public availability of model tools in the field of machine learning have made these techniques easily available to more researchers in the field of geography and ecology (Reichstein et al., 2019). Since the above two major advances (i.e., increasing availability of flux data and machine learning techniques) in the last two decades, various machine learning algorithms have been used to simulate NEE at the flux station scale with various predictor variables (e.g., meteorological variables, biophysical variables) incorporated for spatial and temporal mapping of NEE or understanding the driving mechanisms of NEE.

To date, studies on using machine learning to predict NEE have a high diversity in terms of modeling approaches. To obtain a comprehensive understanding of machine learning-based NEE prediction, a synthesis evaluation of these machine learning models is necessary. Since the beginning of this century, when machine learning approaches were still rarely used in geography and ecology research, neural networks were already used to perform simulations and mapping of NEE in European forests (Papale and Valentini, 2003). Subsequently, considerable efforts have been made by researchers to improve such predictive models. Many studies have demonstrated the effectiveness of their proposed improvements

(i.e., using predictors with a higher spatial resolution (Reitz et al., 2021) and using data from the local flux site network (Cho et al., 2021)) by comparing with previous studies. However, the improvements achieved in these studies may be limited to smaller areas and specific conditions and may not be generalizable (Cleverly et al., 2020; Reed et al., 2021; Cho et al., 2021). We are more interested in guidelines with universal applicability that improve the model accuracy, such as the selection of appropriate predictors and algorithms under different conditions. Therefore, we should synthesize the results of models applied to different conditions and regions to obtain general insights.

[revised manuscript text omitted]

as described below: chosen:

a) Predictors: Various biophysical variables (Zeng et al., 2020; Cui et al., 2021; Huemmrich et al., 2019) and other meteorological and environmental factors have been used in the simulation of NEE. The most commonly used predictor variables include precipitation (Prec), air temperature (Ta), wind speed (Ws), net/sun radiation (Rn/Rs), soil temperature (Ts), soil texture, soil moisture (SM) (Zhou et al., 2020), vapor pressure deficit (VPD) (Moffat et al., 2010; Park et al., 2018), the fraction of absorbed photosynthetically active radiation (FAPAR) (Park et al., 2018; Tian et al., 2017), vegetation index (e.g., NDVI, EVI), LAI, and evapotranspiration (ET) (Berryman et al., 2018). The predictor variables used vary with the natural conditions and vegetation functional types of the study area. In contrast, in models that include multiple PFTs, some variables that play a significant role in the prediction of each of the multiple PFTs may have higher importance. For example, growing degree days (GDD) may be a more effective variable for NEE of tundra in the northern hemisphere high latitudes (Virkkala et al., 2021), while measured groundwater levels may be important for wetlands (Zhang et al., 2021). Some of these predictor variables are measured at flux stations (e.g., meteorological factors such as precipitation and temperature), while others are extracted from reanalyzed meteorological datasets and satellite remote sensing image data (e.g., vegetation indices). The spatial and temporal resolution of predictors can lead to differences in their relevance to NEE observations. Most measured in situ meteorological factors have a good spatio-temporal match to the observed NEE (site scale, half-hourly scale). However, the proportion of NEE explained by remotely sensed biophysical covariates may depend on their spatial and time scales. For example, the MODIS-based 8 daily NDVI data may better capture temporal variation in the relationship between NEE and vegetation growth than the Landsat-based 16 daily NDVI data. In contrast, the interpretation of NEE by variables such as soil texture and soil organic content (SOC), which do not have temporal dynamic information, may be limited to the interpretation of spatial variability, although they are considered to be important drivers of NEE. Therefore, the importance of variables obtained from NEE simulations based on a data-driven approach may differ from that in process-based models as well as in the actual driving mechanisms. This may be related to the spatial and temporal resolution of the predictors used and the quality of the data. It is necessary to consider the spatio-temporal resolution of the data for the actual biophysical variables used in the different studies in the systematic evaluation of data-driven NEE simulations.

b) The spatio-temporal heterogeneity of data sets, and validation method: The spatio-temporal heterogeneity of the dataset may affect model accuracy. Typically, training data with larger regions, multiple sites, multiple PFTs, and longer spans of years may have a higher degree of imbalance (Kaur et al., 2019; Van Hulse et al., 2007; Virkkala et al., 2021; Zeng et al., 2020). Modeling with unbalanced data (where the difference between the distribution of the training and validation sets is significant even if selected at random) may result in lower model accuracy. To date, the most commonly used methods for validating such models include spatial (Virkkala et al., 2021), temporal (Reed et al., 2021), and random (Cui et al., 2021) cross-validation. The imbalance of data between the training and validation sets may affect the accuracy of the models when using these validation methods. Spatial validation is used to assess the ability of the model to adapt to different regions or flux sites of different PFTs, and a common method is 'leave one site out' cross-validation (Virkkala et al., 2021; Zeng et al., 2020). If the data from the site left out is not covered (or partially covered) by the distribution of the training dataset, the model's prediction performance at that site may be poor due to the absence of a similar type in the training set. Temporal validation typically uses some years of data as training and the remaining years as validation to assess the model's fitness for interannual variability. For a year that is left out (e.g. a special extreme drought year which does not occur in the training set), the accuracy of the model may be limited if there are no similar years (extreme drought years) in the training dataset. K-fold cross-validation is commonly used in random cross-validation to assess the fitness of the model to the spatio-temporal variability. In this case, different values of K may also have a significant impact on the model accuracy. For example, for an unbalanced dataset, the average model accuracy obtained from a 10-fold (K = 10) validation approach is likely to be higher than that of a 3-fold (K = 3) validation approach (Marcot and Hanea, 2021).

c) Machine learning algorithms used: Simulating NEE using different machine learning algorithms may influence the model accuracy, which may be induced by the characteristics of these algorithms themselves and the specific data distribution of the NEE training set. For example, Neural Networks can be used effectively to deal with nonlinearities, while as an ensemble learning method, Random Forests can avoid overfitting due to the introduction of randomness. Therefore, a comprehensive evaluation of this is necessary.

In this study, to evaluate the impacts of predictors use, algorithms, spatial/time scale, and validation methods on model accuracy, we performed a meta-analysis of papers with prediction models that combine NEE observations from flux towers, various predictors, and machine learning for the data-driven NEE simulations. In addition, we also analyzed the causality of multiple features in NEE simulations and the joint effects of multiple features on model accuracy using the Bayesian Network (BN) (a multivariate statistical analysis approach (Pearl, 1985)). The findings of this study can provide some general guidance for future NEE simulations.

[revised manuscript text omitted]

设置了格式: 字体: 10 磅

设置了格式: 字体: 10 磅

设置了格式: 字体: 10 磅
设置了格式: 字体: 10 磅
设置了格式: 字体: 10 磅

in this meta-analysis.

**2.2 Features of  prediction processes evaluated**

~~Typically, the flow of the NEE prediction modeling framework (Fig. 2) based on flux observations and machine learning is as follows: first, half-hourly scale NEE flux observations are aggregated into various time scale NEE data, and gap filling techniques (Moffat et al., 2007) are often used in this step to obtain complete NEE series when data are missing. Various predictors including meteorological variables, remote sensing-based biophysical variables, etc. are extracted to match site-scale NEE series to generate a training dataset containing the target variable NEE and various covariates. Subsequently, various algorithms are used for the NEE prediction model construction and validated in different ways (e.g., leave-one-site-out validation (Zeng et al., 2020)). Finally, in some studies, prediction models were applied to gridded covariate data to map the regional or global-scale NEE spatial and temporal variations (Zeng et al., 2020; Papale and Valentini, 2003; Jung et al., 2020). The information of R-squared (at the validation phase) and the associated model features reported in the article are considered as one data record for the formal meta-analysis (i.e., each R-squared record corresponding to a prediction model). From the included papers, R-squared records and various features (Table 2) involved in the NEE modeling framework (Fig. 2) were extracted (including the used algorithms, modeling/validation methods, remote sensing data, meteorological data, biophysical data, and ancillary data). In some studies, multiple algorithms were applied to the same dataset, or models with different features were developed (Virkkala et al., 2021; Zhang et al., 2021; Cleverly et al., 2020; Tramontana et al., 2016). In these cases, multiple data records will be documented.~~

 The various features (Table 2) involved in the water flux modeling framework (Fig. 1) include the PFTs of the sites, the predictors used, the

machine learning algorithms, the validation methods, and other features. Each model for which R-squared is reported is treated as a data record. If multiple algorithms were applied to the same dataset, then multiple records were extracted. Models using different data or features are also recorded as multiple records.

Random Forests (RF), Multiple Linear Regressions (MLR), Artificial Neural Networks (ANN), Support Vector Machines (SVM), Partial Least Squares Regression (PLSR), Generalized additive model (GAM), Boosted Regression Tree (BRT), Bayesian Additive Regression Trees (BART), Cubist, model tree ensembles (MTE). Second, we classified the spatial scales of these studies. Models with study areas (spatial extent covered by flux stations) smaller than 100x100 km were classified as 'local' scale models, those with study area sizes exceeding continental scale were classified as 'global' scale, and those with study area sizes in between were classified as 'regional' scale. Third, for various predictors, we only recorded whether the predictors were used or not without distinguishing the detailed data sources and categories (e.g., grid meteorological data from various reanalysis datasets and in-situ meteorological observations from flux stations), measurement methods (e.g., soil moisture measured/estimated by remote sensing or in situ sensors), etc. Fourth, we documented PFTs for the prediction models from the description of study areas or sites in these papers. They are classified into the following types: forest, grassland, cropland, wetland, savannah, tundra, and multi-PFTs (models containing a mixture of multiple PFTs). Models not belonging to the above PFTs were not given a PFT field and were not included in the subsequent analysis of the PFT differences. Other features (Table 2) are extracted directly from the corresponding descriptions in the papers in an explicit manner.

Subsequently, the model accuracies corresponding to different levels of various features are compared in a cross-study fashion. In the evaluation of algorithms and time scales, we also implement comparisons within individual studies. For example, in the evaluation of the effects of the algorithms, we compare the accuracy of models using the same training data and keeping other features as constants in individual studies. In this intra-study comparison step, only algorithms with relatively large sample sizes in the cross-study comparisons were selected. In this study, algorithms with less than 10 available model records are not considered to have a sufficient sample size and we do not give further conclusive opinions on the accuracy of these algorithms due to their small samples (e.g., PLSR and BART with high R-

squared but very few records as evidence). MLR, RF, SVM, and ANN were found to have large sample sizes (Fig. 5a), and thus their accuracies can be comparable. Based on this, in the intra-study comparison step, we only compare the accuracy differences between MLR, RF, SVM, and ANN in the context of using the same data and the same other model features (Fig. 5b).

Figure 2. Features of the machine learning-based NEE prediction process. The flux tower photo is from https://www.licor.com/env/support/Eddy-Covariance/videos/ec-method-02.html (last accessed: 23rd March 2022). The map in the lower part is from Harris et al., 2021. Prec, Ta, Rn, Ws, RH, and VPD represent precipitation, air temperature, net surface radiation, wind speed, relative humidity, and vapour-pressure deficit respectively. FAPAR is the fraction of absorbed photosynthetically active radiation. LST is the land surface temperature. LAI is the leaf area index.

**2.3 Bayesian Network for analyzing joint effects**

Based on the Bayesian network (BN), the joint impacts of multiple model features on the R-squared are analyzed. A BN can be represented by nodes ($X_1,.., X_n$) and the joint distribution (Pearl, 1985):

$$P(X) = P(X_1, X_2, ..., X_n) = \prod_{i=1}^{n} P(X_i|pa(X_i)) \tag{1}$$

where $pa(X_i)$ is the probability of the parent node $X_i$. Expectation-maximization (EM) approach (Moon, 1996) is used to incorporate the collected model records and compile the BN.

Sensitivity analysis is used for the evaluation of node influence based on mutual information (MI) which is calculated as the entropy reduction of the child node resulting from changes at the parent node (Shi et al., 2020):

$$MI = H(Q)-H(Q|F) = \sum_q \sum_f P(q, f) \log_2\left(\frac{P(q,f)}{P(q)P(f)}\right) \tag{2}$$

where H represents the entropy, Q represents the target node, F represents the set of other nodes and q and f represent the status of Q and F.

**3 Results**

**3.1 Articles included in the meta-analysis**

We included 40 articles (Table S2) and extracted 178 model records for the formal meta-analysis (Fig. 1). Most studies were implemented in Europe, North America, Oceania, and China (Fig. 3). The number of such papers is increasing recently (Fig. 4) and it shows the machine learning approach for NEE prediction has been of interest to more researchers. The main journals in which these articles have been published (Fig. 4) include Remote Sensing of Environment, Global Change Biology, Agricultural and Forest Meteorology, Biogeosciences, and Journal of Geophysical Research: Biogeosciences, etc.

Figure 3. Location of studies (a) included with the number of flux sites included and (b) their PFTs in the meta-analysis (total of 40 studies and 178 model records). Global (mainly based on FluxNet (Tramontana et al., 2016)) and continental-scale studies are not shown on the map due to the difficulty of identifying specific locations.

Figure 4. The number of studies published across journals and the total number of publications per year.

A total of 32 articles (Table S1) containing a total of 139 model records were included. The geographical scope of these articles was mainly Europe, North America, and China (Fig. 2).

**3.2 The formal Meta-analysis**

We assessed the impact of the features (e.g., algorithms, study area, PFTs, amount of data, validation methods, predictor variables, etc.) used in the different models based on differences in R-squared.

**3.2.1 Algorithms**

Among the more frequently used algorithms, ANN SVM and SVM performed betterRF outperformed (Fig. 5a) on average3a) across studies (lightly better than RF). On the other hand, since cross-study comparisons of algorithm accuracy include differences in data used in

model construction, we performed a pairwise comparison (Fig. 5b) of these four  These three machine learning algorithms (i.e., ANN, SVM, RF) were significantly more accurate than the traditional MLR. Other algorithms such as MTE, ELM, Cubist, etc. also correspond to high accuracy, but with limited evidence sample size (Fig. 3a). In the internal comparison (different algorithms applied to the same data set) in single studies, we also find that  SVM and RF were slightly more accurate than ANN (Fig. 3b), and all these three (i.e., ANN, SVM, RF) are considerably more accurate than MLR. Overall,  SVM and RF have shown higher accuracy in water flux simulations in both inter and intra-study comparisons with sufficient sample size as evidence.

**3.2.2 Climate types and PFTs**

We found higher average model accuracy in arid climate zones (Fig. 4a), such as the Cold semi-arid (steppe) climate (BSk) and Cold desert climate (BWk). Most of these studies were located in northwest China and the western USA. It may be caused by the simpler relationship between water fluxes and biophysical covariates in arid regions. In arid zones, due to the high potential ET, the variability in the actual ET may be largely explained by water availability (moisture supply) and vegetation change with the effect of variability in thermal conditions reduced. As for the various PFTs, the average model accuracy was slightly lower for forest types than for cropland and grassland types (Fig. 4b). The lowest average accuracy was found for shrub sites, which may be related to the difficulty of the remote sensing-based vegetation index (e.g., NDVI) to quantify the physiological and ecological conditions of shrubs (Zeng et al., 2022), and the heterogeneity of the spatial distribution of shrubs within the EC observation area may also cause difficulties in capturing their relationships with biophysical variables. We also found high model accuracy for the wetland type, although records as evidence to support this finding may be limited. Compared to other PFTs, the more steady and adequate water availability in the wetland type may make the variations of water fluxes less explained by other biophysical covariates.

**3.3.3 Predictors and their combinations**

On one hand, for the effects of individual predictors, the use of Rn/Rs, Prec, Ta, and FAPAR improved the accuracy of the model (Fig. S1). This pattern partially changed in the different PFTs. In the forest sites, the accuracy of the models with Rn/Rs and Ta used was higher than that of the models with Rn/Rs and Ta not used. For the grassland sites, the use of Ws, FAPAR, Prec, and Rn/Rs improved the model accuracy. For the cropland sites, Ta and FAPAR were more important for improving the model accuracy.

On the other hand, the evaluation of the effect of individual predictors on model accuracy is not necessarily reliable because some predictor variables are used together (e.g., the high model accuracy corresponding to a particular variable may be because it is often used together with another variable that plays the dominant role in improving accuracy). Therefore, we tested for independence between the use of variables and assessed the effect of the combination of variables on model accuracy. We calculated the correlation matrix (Fig. S2) between the use of various predictors (not used is set as 0 and used is set as 1). We found there was a dependence between the use of some predictors, the use of NDVI/EVI, LAI, and SM was significantly negatively correlated with the use of Rn/Rs and Ta (Fig. S2). ~~The impact of time scale on R-squared is considerable (Fig. 6), with models with larger time scales having lower average R-squared, especially when the time scale exceeds the monthly scale. The most frequently used scales were the daily, 8-day, and monthly scales. In studies where multiple time scales were used with other characteristics being the same, we found that models with half-hourly scales were significantly more accurate than models with daily scales (Fig. 6). However, the difference in accuracy between the day-scale and week-scale models is small. The accuracy of models with a monthly scale is the lowest.~~

~~Figure 6. Differences in model accuracy (R-squared) at different time scales across studies with the linear regression between R-squared and time scales (a), and comparison of the model accuracy (R-squared) of selected pairs of time scales within individual studies (b). All model records were included in panel (a), while studies that used multiple time scales (with other model characteristics unchanged) were included in panel (b). Time scales: 0.02 days (half-hourly), 0.04 days (hourly), 30 days (monthly), and 90 days (quarterly).~~

It indicated that many of the models that used Rn/Rs and Ta did not use

[revised manuscript text omitted]

Among the commonly used predictors for NEE, there are significant differences in the predictors used and their impacts on model accuracy for different PFTs (Fig. 7). Ancillary data (e.g. soil texture, soil organic content, topography) that do not have temporal variability are used less frequently because they can only explain spatial heterogeneity. In contrast, the biophysical variables LAI, FAPAR, and ET were used significantly less frequently than NDVI/EVI, especially in the cropland and wetland types. The meteorological variables Ta, Rn/Rs, and VPD were used most frequently. For forest sites, Rn/Rs and Ws

appear to be the variables that improve model accuracy. For grassland sites, we found that NDVI/EVI appears to be the most effective, despite the small sample size. For sites in croplands and wetlands, we did not find predictor variables that had a significant impact on model accuracy.

For different PFTs, the top three variables in the ranking of model importance differed (Fig. S1). SM, Rn/Rs, Ta, Ts, and VPD all showed high importance across PFTs. This suggests that the variability of measured site-scale moisture and temperature conditions is important for the simulation of NEE for all PFTs. In contrast, in the importance ranking, other variables such as precipitation and NDVI/EVI may not lead because of the lag in their effect on NEE (Hao et al., 2010; Cranko Page et al., 2022). And some other variables may improve model accuracy for specific PFTs such as groundwater table depth (GWT) for wetland sites and growing degree days (GDD) for tundra sites.

Figure 7. The impact of the various predictors incorporated in models of different PFTs (1-forest, 2-grassland, 3 cropland, 4 wetland, 6 tundra) on R squared. Dark blue boxes indicate that the predictor was used in the model, while dark red boxes indicate that the predictor was not used. Predictors: soil organic content (Soil_OC), precipitation (Prec), soil moisture/land surface water index (SM_LSWI), net radiation/solar radiation (Rn_Rs), enhanced vegetation index (EVI), air temperature (Ta), vapor pressure deficit (VPD), the fraction of absorbed photosynthetically active radiation/photosynthetically active radiation (FAPAR_PAR), relative humidity (RH), evapotranspiration (ET), leaf area index (LAI).

**3.2.4 Other features**

In addition, we evaluated other features of the model construction that may contribute to differences in model accuracy (Fig. 8). Studies at continental and global scales with a large number of sites and a large span of years correspond to lower R-squared than studies at local and regional scales, suggesting that studies with a large number of sites across large regions are likely to have high variability in the relationship between NEE and covariates and that studies at small scales are more likely to have higher model accuracy. Spatial validation (usually 'leave-one-site-out') corresponds to lower model accuracy compared to random and temporal validation. This again confirms the dominant role of heterogeneity in the relationship between NEE and covariates across sites in explaining model accuracy. This seems to

be indirectly supported by the fact that a high ratio of training to validation sets corresponds to a low R-squared, as this high ratio tends to be accompanied by the use of the 'leave one site out' validation approach. The accuracy of the models with a growing season period was slightly higher than that of the models with an annual period. For the satellite remote sensing data used, the models based on MODIS data with biophysical variables extracted were slightly less accurate than those based on Landsat data. For the daily scale models, Landsat data performed a little better than MODIS (Fig. S2). This suggests that the higher temporal resolution of MODIS compared to Landsat may not play a dominant role in improving model accuracy. This may also be partially attributed to studies using MODIS based explanatory data that tend to include too large surrounding areas around the site (e.g., 2x2 km), which can lead to a scale mismatch between the flux footprint and the explanatory variables.

Figure 8. The impacts of other features (i.e. spatial scale, study period, number of sites, year span, site year, cross-validation method, training/validation, and satellite imagery) on the model performance.

**3.3. The joint causal impacts of multi-features based on the BN**

We selected the features that had a more significant impact on model accuracy in the above assessment and further incorporated them into the BN-based multivariate assessment to understand the joint impact of multiple features on R-squared. The features incorporated included the spatial scale, the number of sites, the time scale, the span of years, the cross-validation method, and whether some specific predictors were used. We discretized the distribution of individual nodes and compiled the BN (Fig. 9a) using records from different PFTs as input. Sensitivity analysis of the R-squared node (Fig. 10) showed that R-squared was most sensitive to 'year span', cross-validation method, Rn/Rs, and time scale under multi-feature control. In the forest and cropland types, R-squared is more sensitive to Rn/Rs, while in the wetland type it is more sensitive to SM/LSWI and Ta. The sensitivity of R-squared to 'year span' was much higher in the cropland type compared to the other PFTs, which may suggest that the interannual variability in the NEE simulations of the cropland type is higher due to potential interannual variability of the planting structure and irrigation practices. For the cropland type, differences in the phenology, harvesting, and irrigation (water volume and frequency) in different years can lead to significant inter-annual differences in NEE simulations. Subsequently, using the constructed BN (with the empirical information in previous studies incorporated), for new studies we can instructively infer the probability

distribution of the possible R-squared (Fig. 9b) with some model features predetermined. In previous studies, spatio-temporal mapping of NEE based on statistical models has often lacked accuracy assessment since there are no grid-scale NEE observations, and this BN may have the potential to be used to validate the accuracy (R-squared) of the NEE time series output of the grid scale (i.e. inferring possible R-squared from model features, where the output of the grid-scale is considered to be of the form 'leave one site out').

Figure 9. The joint effects of multiple features on the R-squared based on the BN with all records input (a) and the inference on the probability distribution of R-squared based on the BN with the status of some nodes determined (b). The values before and after the "±" indicate the mean and standard deviation of the distribution, respectively. The gray boxes indicate that the status of the nodes has been determined. In panel (b), specific values of parent nodes such as 'spatial scale' are determined (shown in the red box), leading to an increase in the expected R-squared compared to the average scenario of panel (a) (as inferred from the posterior conditional probabilities with the status of the node 'spatial scale' are determined as 'local').

Figure 10. The sensitivity analysis of the R-squared node to other nodes based on the mutual information (MI) across PFTs. 'Cross validation' is the cross-validation method including spatial, temporal, and random cross-validation.

**4 Discussions**

[revised manuscript text omitted]

Among the Many studies have evaluated the incorporation of various predictors and model features using machine learning for improving the site-scale NEE predictions (Tramontana et al., 2016; Zeng et al., 2020; Jung et al., 2011). A comprehensive evaluation of these studies to provide definitive guidance on the selection of features in NEE prediction modeling is limited. This study fills the research gap with a meta-analysis of the literature through statistics on the accuracy and performance of models. Machine

learning-based NEE simulations and predictions still suffer from high uncertainty. By better understanding the expected improvements that can be achieved through the inclusion of different features, we can identify priorities for the consideration of different features in modeling efforts and avoid operations decreasing model accuracy.

Compared to previous comparisons of machine learning-based NEE prediction models, this study is more comprehensive. Previous studies (Abbasian et al., 2022) have also found advantages of RF over other algorithms in NEE prediction. This study consolidated this finding using a larger amount of evidence. Previous studies (Tramontana et al., 2016) have also compared the impact of different practices in NEE prediction models based on the R-squared, such as comparing the difference in accuracy between the two predictor combinations (i.e., using only remotely sensed data and using remotely sensed data and meteorological data together). In contrast, since this study incorporated more detailed factors influencing model accuracy, the understanding of such issues was deepened. However, there are still many uncertainties and challenges in NEE prediction not clarified in this study.

**4.1 Challenges in the site-scale NEE simulation and implications for other carbon flux simulations**

**4.1.1 Variations in time scales**

In the above analysis, we found that the effect of the time scale of the model is considerable. This suggests that we should be careful in determining the time scale of the model to consider whether the predictor variables used will work at this time scale. Previous studies have reported the dependence of the NEE variability and mechanism on the time scales. On the one hand, the importance of variables affecting NEE varies at different time scales. For example, in tropical and subtropical forests in southern China (Yan et al., 2013), seasonal NEE variability is predominantly controlled by soil temperature and moisture, while interannual NEE variability is controlled by the annual precipitation variation. A study (Jung et al., 2017) showed that for annual-scale NEE variability, water availability and temperature were the dominant drivers at the local and global scales, respectively. This indicates the need to recognize the temporal and spatial driving mechanisms of NEE in advance in the development of NEE prediction models. On the other hand, dependence may exist between NEE anomalies at various time scales. For example, previous studies (Luyssaert et al., 2007) showed that short-term temperature anomalies may interpret both the daily and seasonal NEE anomalies. This implies that the models at different time scales

may not be independent. In the previous studies, the relationship between prediction models at different scales has not been well investigated, and it may be valuable to compare the relations between data and models at different scales in depth. Larger time scales correspond to lower model accuracy, possibly related to the fact that some small time scale relations between NEE and covariates (especially meteorological variables) are smoothed. In particular, for models with time scales smaller than one day (e.g. half hourly models), the 8-daily and 16-daily biophysical variable data obtained from satellite remote sensing are difficult to explain the temporal variation in the sub-daily NEE. Therefore, for models at small time scales (i.e. half hourly, hourly, daily scale models), in situ meteorological variables may be more important. The inclusion of some ancillary variables (e.g. soil texture, topographic variables) with no temporal dynamic information may be ineffective unless many sites are included in the model and the spatial variability of the ancillary variables for these sites is sufficiently large (Virkkala et al., 2021).

In terms of completeness and purity of training data, hourly and daily models can be better compared to monthly and yearly models. Hourly and daily models can usually preclude those low-quality data and gaps in the flux observations. However, for monthly and yearly scale models, gap filling (Ruppert et al., 2006; Moffat et al., 2007; Zhu et al., 2022) is necessary because there are few complete and continuous fluxes observations without data gaps on the monthly to yearly scales. Since various gap filling techniques rely on environmental factors (Moffat et al., 2007) such as meteorological observations, this may introduce uncertainty in the predictive models (i.e., a small fraction of the observed information of NEE is estimated from a combination of independent variables). How it would affect the accuracy of prediction models at various time scales remains uncertain, although various gap filling techniques have been widely used in the pre-processing of training data.

In addition, the impacts of lagged effects (Hao et al., 2010; Cranko Page et al., 2022) of covariates are not considered in most models, which may underestimate the degree of explanation of NEE for some predictor variables (e.g. precipitation). Most of the machine learning-based models use only the average Ta and do not take into account the maximum temperature, minimum temperature, daily difference in temperature, etc., as in the process-based ecological models (Mitchell et al., 2009). This suggests that the inclusion of different temporal characteristics of individual variables in machine learning-based NEE prediction models may be insufficient.

**4.1.2 Scale mismatch of explanatory predictors and flux footprints**

An excessively large extraction area of remote sensing data (e.g., 2x2 km) may be inappropriate. In the non-homogeneous underlying conditions, the agreement of the area of flux footprints with the scale of the predictors should be considered in the extraction of the predictor variables in various PFTs (Chu et al., 2021).

The effects of this mismatch between explanatory variables and flux footprints may be diverse for different PFTs. For example, for cropland types, the NEE is monitored at a range of several hundred meters around the flux towers, but remote sensing variables such as FAPAR, NDVI, LAI, etc. can be extracted at coarse scales (e.g., 2x2 km), some effects outside the extent of the flux footprint (Chu et al., 2021; Walther et al., 2021) are incorporated (e.g., planting structures with high spatial heterogeneity, agricultural practices such as irrigation). And for more homogeneous types such as grasslands, coarse-scale meteorological data may still cause spatial mismatches, even though the differences in land cover types within the 2x2 km and 200x200 m extent around the flux stations in grasslands may not be considerable. For example, precipitation with high spatial heterogeneity can dominate the spatial variability of soil moisture and thus affect the spatial variability of grassland NEE (Wu et al., 2011; Jongen et al., 2011). However, using 0.25°x0.25° reanalysis precipitation data (Zeng et al., 2020) may make it difficult for predictive models to capture this spatial heterogeneity around the flux station.

Since few of the studies included in this meta-analysis considered the effect of variation in flux footprint, this feature was difficult to consider in this study. However, its influence should still be further investigated in future studies. With flux footprints calculated (Kljun et al., 2015) and the factors around the flux site (Walther et al., 2021) that affect the flux footprint incorporated, .it is promising to clarify this issue.

**4.1.3 Possible unbalance of training and validation sets**

In addition to the time scale of the models, the most significant differences in model accuracy and performance were found in the heterogeneity within the NEE dataset and the match of the training set and validation set. Often NEE simulations can achieve high accuracy in local studies, where the main factor negatively affecting model accuracy may be the interannual variability in the relationship between

NEE and covariates. However, the complexity may increase when the dataset contains a large study area, many sites, PFTs, and year spans. Under this condition, the accuracy of the model in the 'leave one site out' validation may be more dependent on the correlation and match between the training and validation sets (Jung et al., 2020). When the model is applied to an outlier site (of which the NEE, covariates, and their relationship are very different compared with the remaining sites), it appears to be difficult to achieve a high prediction accuracy (Jung et al., 2020). If we further upscale the prediction model to large spatial and time scales, the uncertainties involved may be difficult to assess (Zeng et al., 2020). We can only infer the possible model accuracy based on the similarity of the distribution of predictors in the predicted grid to that of the existing sites in the model. In the upscaling process, reanalysis data with coarse spatial resolution are often used as an alternative for site-scale meteorological predictors. However, most studies did not assess in detail the possible errors associated with spatial mismatches in this operation.

In summary, the site-scale NEE predictions may require more focus on the internal heterogeneity of the NEE dataset and the matching of the training set and validation set, and also require a better understanding of the influence of different scales of the same variable (e.g. site-scale precipitation and grid-scale precipitation in the reanalysis meteorological data) across modeling and upscaling steps. For the prediction of other carbon fluxes such as methane fluxes (in the same framework as the NEE predictions), the results of this study may also be partially applicable, although there may be significant differences in the use of specific predictors (Peltola et al., 2019).

**4.2 Uncertainties**

The uncertainties in this analysis may include:

[revised manuscript text omitted]

**4.2.2 Publication bias and weighting:**

a) Publication bias is not refined dueand weighting: Due to the limitations of the relatively limited number of articles that cancould be included in the meta-analysis, this study did not focus much on publication bias. Meta-analyses oftenanalytic studies in other fields typically measure the quality of journals and the datapublic availability of research data (Borenstein et al., 2011; Field and Gillett, 2010) to determine the weighting of the literature in a comprehensive assessment. However, a high proportionmost of the articles in this study did not makepublicly provide flux observations publicly available or share the NEE prediction models developed. Furthermore, meta models. Meta-analysis studies in other fields typically measure the impact of papers by evidence/data volume;included studies based on sample size and the variance of the evaluated

experimental results (Adams et al., 1997; Don et al., 2011; Liu et al., 2018a). In this study, due to the lack of a convincing manner to determine weights among articles, we assigned the same weight to the results for all the literature.

~~b)   Limitations of the criteria for inclusion in the literature: in the model accuracy-based evaluation, we selected only literature that developed multiple regression models. Potentially valuable information from univariate regression models was not included. In addition, only papers in high-quality English journals were included in this study to control for possible errors due to publication bias. However, many studies that fit this theme may have been published in other languages or other journals.~~

~~c)   Independence between features: There is dependence between the evaluated features (e.g. the dependency between the spatial extent and the number of sites). It may negatively affect the assessment of the impact of individual features on the accuracy of the model, although the BN-based analysis of joint effects can reduce the impact of this dependence between variables by specifying causal relationships between features.  The interference of unknown dependencies between features may still not be eliminated when we focus on the effects of an individual feature on the model performance. We should pay more attention to the effect of features on model accuracy individually in future studies, and it may be valuable to keep other features as constants while changing the level of only one feature and assessing the difference. It may help us to understand the real sensitivity of model accuracy to different features in specific conditions. The sample size collected in this study (178 records in total) is not very large. This also suggests that more future efforts should be devoted to the comprehensive evaluation and summarization of NEE simulations.~~

设置了格式: 字体: 10 磅

设置了格式: 字体: 10 磅

**4.2.3 Uncertainties in the information of the extracted features**

At the information extraction level, the following issues may also introduce uncertainties:

a)  Uncertainties caused by data quality control (e.g. gap-filling (Hui et al., 2004)) are difficult to assess effectively. Gap-filling is a commonly used technique to fill in low-quality data in flux observations. However, the impact of this practice on machine learning-based ET prediction models is unclear, due to the difficulty of directly assessing how this technique is performed in various studies by this meta-analysis. Typically, models with small time scales (e.g., hourly scale, daily scale) can exclude low-quality observations and use only high-quality data. However, for models with large time scales (e.g., monthly scales), gap-filling (e.g., based on meteorological data) may be unavoidable. This may lead to a decrease in training data purity and introduce uncertainty in the subsequent prediction model development.

b)  Systematic uncertainties caused by the energy balance closure (EBC) issue in eddy-covariance flux measurements are also difficult to assess by this meta-analysis. EBC is a common problem (Eshonkulov et al., 2019) in eddy-covariance flux observations. For that reason, the latent heat flux measured potentially underestimates ET. Some prediction models corrected EBC (e.g., using Bowen ratio preserving (Mauder et al., 2013, 2018) and energy balance residuals (Charuchittipan et al., 2014; Mauder et al., 2018)) in the processing of training data, but some did not. How this will affect the accuracy of the prediction model is not clear due to multiple factors that need to be evaluated that influence EBC (Foken, 2008), including measurement errors of the energy balance components, incorrect sensor configurations, influences of heterogeneous canopy height, unconsidered energy storage terms in the soil-plant-atmosphere system, inadequate time averaging intervals, and long-wave eddies (Jacobs et al., 2008; Foken, 2008; Eshonkulov et al., 2019). To reduce this uncertainty, more attention to flux site characteristics (Eshonkulov et al., 2019) related to PFT, topography, flux footprint area, etc., to select the appropriate correction method is necessary for future studies.

c)  As most studies used far more water flux observation records than the number of covariates in their regression models, we did not adjust the R-squared in this study to an adjusted R-squared.

d)  The various specific ways in which the parameters of the model are optimized are not differentiated. They are broadly categorized into different families or kinds of algorithms, which may also introduce uncertainty into the assessment.

e) The assessment of some features is not detailed due to the limitations of the available model records. For example, the classification of PFT could be more detailed. 'Forest' could be further classified as broadleaf forest, coniferous forest, etc. while 'cropland' could be further classified as rainfed and irrigated cropland based on differences in their response mechanisms of water fluxes to environmental factors.

**5 Conclusion**

We performed a meta-analysis of the  water flux simulations combining in situ flux observations from flux stations/networks, meteorological, biophysical, and ancillary predictors, and machine learning.  The main  conclusions are as follows:

1.  SVM (average R-squared = 0.82) and

 RF (average R-squared = 0.

1. 81) outperformed over evaluated algorithms with sufficient sample size in  both cross-study and  intra-study (with the same training dataset) comparisons.

2.  The average accuracy of the model applied to arid regions is  higher than in other climate types.

3. The average accuracy of the model was slightly lower for forest sites (average R-squared = 0.76) than for cropland and grassland sites (average R-squared  = 0.8 and 0.79), but higher than for shrub sites (average R-squared = 0.67).

4. Among various predictor variables, the use of Rn/Rs, Prec, Ta, and FAPAR improved the model accuracy. The combination of Ta and  Rn/Rs is very effective especially in the forest type, while in the grassland type the combination of Ws and Rn/Rs is also effective.

**Acknowledgments**

5. Among the different validation methods, random cross-validation shows higher model accuracy than spatial cross-validation and temporal cross-validation.

**Acknowledgments**

**Acknowledgements**

We thank the editor and two anonymous reviewers for their insightful comments  which contributed substantially to the improvement of this manuscript.

**Financial support**

This research was supported by the National Natural Science Foundation of China (Grant No. U1803243), the Key projects of the Natural Science Foundation of Xinjiang Autonomous Region (Grant No. 2022D01D01), the Strategic Priority Research Program of the Chinese Academy of Sciences (Grant No. XDA20060302), and High-End Foreign Experts Project.

**Author  Contributions**

HS and  GL were responsible for the conceptualization, methodology, formal analysis, investigation, visualization, and writing. OH contributed to the investigation. XM, XY, YW, WZ, MX, CZ and YZ processed the data. AK, TVDV and PDM provided supervision.

**Competing interests**

The authors declare that they have no conflict of interest.

**Code availability**

The codes that were used for all analyses are available from the first author (shihaiyang16@mails.ucas.ac.cn) upon request.

**Data availability**

The data used in this study can be accessed by contacting the first author (shihaiyang16@mails.ucas.ac.cn) upon request.

设置了格式: 字体: 10 磅

